# Defects and Defect Engineering of Two-Dimensional Transition Metal Dichalcogenide (2D TMDC) Materials

**DOI:** 10.3390/nano14050410

**Published:** 2024-02-23

**Authors:** Moha Feroz Hossen, Sachin Shendokar, Shyam Aravamudhan

**Affiliations:** 1Joint School of Nanoscience and Nanoengineering, 2907 E Gate City Blvd, Greensboro, NC 27401, USA; mhossen2@aggies.ncat.edu (M.F.H.); smshendo@aggies.ncat.edu (S.S.); 2Department of Nanoengineering, North Carolina Agricultural and Technical State University, Greensboro, NC 27411, USA

**Keywords:** transition metal dichalcogenides, 2D materials, defects, grain boundaries, characterization tools, heterostructures, electronic and optical properties

## Abstract

As layered materials, transition metal dichalcogenides (TMDCs) are promising two-dimensional (2D) materials. Interestingly, the characteristics of these materials are transformed from bulk to monolayer. The atomically thin TMDC materials can be a good alternative to group III–V and graphene because of their emerging tunable electrical, optical, and magnetic properties. Although 2D monolayers from natural TMDC materials exhibit the purest form, they have intrinsic defects that limit their application. However, the synthesis of TMDC materials using the existing fabrication tools and techniques is also not immune to defects. Additionally, it is difficult to synthesize wafer-scale TMDC materials for a multitude of factors influencing grain growth mechanisms. While defect engineering techniques may reduce the percentage of defects, the available methods have constraints for healing defects at the desired level. Thus, this holistic review of 2D TMDC materials encapsulates the fundamental structure of TMDC materials, including different types of defects, named zero-dimensional (0D), one-dimensional (1D), and two-dimensional (2D). Moreover, the existing defect engineering methods that relate to both formation of and reduction in defects have been discussed. Finally, an attempt has been made to correlate the impact of defects and the properties of these TMDC materials.

## 1. Introduction

The discovery of graphene generated interest among researchers regarding layered materials. Transition metal dichalcogenides (TMDs) with a layered structure have attracted significant attention in the extensive exploration of two-dimensional (2D) materials [1]. Naturally, materials may contain defects, and defects are invariably induced in synthetic materials for a multitude of factors, such as the process kinetics and the tendency of any system to gain lower surface energy pertaining to thermal equilibrium [2]. Defects have functional importance when it comes to semiconductors because the electrical, optical, and magnetic properties of semiconductors are modulated by tuning defect density. Indeed, some applications require being defect-free, and others require artificially created defects in the semiconductors. However, in next-generation channel material, ultra-thin semiconductors are needed because they minimize the present limitations of nanodevices, and new physics emerge due to the reduction in dimensions. Indeed, two-dimensional (2D) layered-crystal materials such as graphene, transition metal dichalcogenides (TMDCs) [3], black phosphorous (BP) [4], hexagonal boron nitride (h-BN) [5], rhenium disulfide (ReS_2_) [6], and rhenium diselenide (ReSe_2_) [7] are emerging materials with novel physical properties such as the valley effect, excitonic fluidity, topologically protected states, and moiré physics used in next-generation nanoelectronics [1]. Among various 2D materials, graphene is widely used. However, its zero bandgap switches the focus of researchers to other materials like TMDC materials, and the band gap of TMDC materials within the visible to near-infrared region facilitates remarkable properties, including electrostatic coupling, photo-switching, gate-tunable superconductivity, and valleytronics [8]. Additionally, the carrier mobility of TMDC materials is comparatively higher than graphene [8]. Moreover, the properties of the TMDC materials depend on the layers’ thickness, making these TMDC materials promising ones for electronics and optoelectronics applications. The various synthesis techniques of TMDC materials have been applied, such as mechanical exfoliation, liquid exfoliation, chemical vapor deposition (CVD), physical vapor deposition (PVD) [9], atomic layered deposition (ALD) [10], chemical vapor transport (CVT) [11], molecular beam epitaxy (MBE) [12], and metal–organic chemical vapor deposition (MOCVD) [13,14]. Indeed, each synthesis process generates different types of defects in these materials [9]. There are several reasons for such defects in these materials, including the absence of either transition metal atoms or chalcogen atoms, substitutional atoms, adatoms, grain boundaries, charged impurities, oxidations, and vdW hetero-epitaxy. These defects are divided into zero-dimensional (0D), one-dimensional (1D), and two-dimensional (2D) [2]. However, these defects are either intrinsic or extrinsic depending on the source of the defects. These defects can modulate the electronic, optical, and magnetic properties of the TMDC materials. For example, anti-site defects in MoS_2_ change the electronic band structure by splitting d-orbital energies and create triplet and singlet states inside the bandgap. Such anti-site defects act as quantum bits [15]. Also, the formation of chalcogen vacancies in the pristine MoS_2_ generates intermediate energy states that are suitable for photonic emission [16]. Additionally, anti-site defect formation during growth in MoTe_2_ changes the magnetic properties of such materials [17]. Moreover, point defects, especially chalcogen vacancies in TMDC materials, reduce the contact resistance at the metal–TMDC interface [18]. Also, overlapping or merging of grain boundaries of TMDC materials impact both the electrical and optical properties. Both pristine and defect-induced 2D TMDC materials have specific applications. Therefore, controlling defects is crucial in TMDC materials. The formation or mitigation of defects, either by healing or formation, of the TMDC materials can be performed in two ways: in situ and ex situ processes [2].

This review focuses on various defects in transition metal dichalcogenides, various defect characterization tools, the impacts of defects on electronic, optical, and magnetic properties of materials, and the defect engineering methods both for the generation and mitigation of the defects in the TMDC materials.

## 2. Fundamental Properties of the TMDC Materials

Transition metal dichalcogenides are layered materials, and the general formula of these materials is MX_2_, where M represents transition metal atoms (e.g., Mo, W, Ti, Nb, Sc, Pd, Pt, V, Mn, and Ru) and X represents chalcogen atoms (e.g., S, Se, and Te). The sandwich structure X–M–X of each 2D layer is covalently bonded, where each 2D layer with a thickness of ~6.50 A is vertically stacked over one another by weak van der Waals forces, as shown in Figure 1a. In each sandwich layer, the charges of transition metal and chalcogen atoms are (+4) and (−2), respectively. Additionally, the lone pair electrons of SP3 hybridized chalcogen atoms leave the surface and create perpendicular, trigonal, or hexagonal symmetry. The TMDC material surface is free from dangling bonds, which facilitates the vertical formation of heterostructures without lattice mismatch [19]. There are several structural phases of TMDC materials existing due to the variation in the coordination number of transition metals. As a matter of fact, the electron’s occupancy of the d-orbital determines the phase structure of the TMDC materials. The two typical structural phases are octahedral symmetry (1T) and trigonal prismatic (2H). In the 2H phase represented in Figure 1b; the atomic stacking is an ABA combination on top of each other where two chalcogen atoms of different planes possess similar positions opposite to the layer. 

On the other hand, the 1T phase has ABC stacking order and octahedral coordination, as depicted in Figure 1b. In this phase, the position of one plane chalcogen atom is shifted from the other. These two phases can be either stable or metastable. For example, the TMDC materials with the 2H phase formed by group 6 transition metals (such as Mo and W) are thermodynamically stable, while the 1H phase is metastable for monolayer TMDC. The exceptional material of this group is WTe_2_, exhibiting a thermodynamically stable state in the 1T phase. Another stacking configuration is the 1T’ phase in which a metal–metal configuration is formed, such as 1T-ReS_2_ [1]. However, the bandgap structure of TMDC materials changes with the thickness of 2D layers. This realization comes from the monolayer MoS_2_ because the bandgap for monolayer (ML) MoS_2_ is 1.8 eV, but the bandgap of this bulk material is 1.2 eV. The reason for this bandgap variation is the transition in the location of the valence band and conduction band edges with the decrease in the number of layers. The accompanying effect is the bandgap transition from indirect to direct, as demonstrated in Figure 2a.

Additionally, this bandgap transition happens because of quantum-mechanical confinement in the z-direction as well as a change in the orbital hybridization of the M and X atoms [1]. This bandgap transition of TMDC materials is confirmed by the photoluminescence (PL) spectroscopy exhibited in Figure 2b. The PL intensity increases with decreasing the number of layers. Moreover, the monolayer 2H TMDC materials show a lack of inversion symmetry, leading to the spin-splitting of the electronic band. It has been identified that these properties are analogous among other TMDC materials, such as MoSe_2_, WSe_2_, MoTe_2_, and WS_2_. While evaluating the binding energy of excitons against the magnitude of bandgap, there happens to be a direct correlation with the optical functionalities. For instance, the monolayer TMDC materials’ refractive index rises with increasing the wavelength of incidence light ranging from 193 nm to 550 nm. As a non-linear optical characteristic, second harmonic generation (SHG) is found in the TMDC materials because they lack inversion symmetry. However, the relation between SHG and the thickness of the TMDC materials is irregular [21]. From the mechanical perspective, the strain limit of TMDC materials is up to 25% compared to graphene [21]. Additionally, there is a linear relationship between the bandgap and mechanical stress of the TMDC materials. When the mechanical stress is increased, the bandgap decreases proportionally and the direct bandgap of monolayer TMDC materials is converted to an indirect bandgap, leading to low emission efficiency. This tunable property by mechanical stress can be used in various flexible electronic devices [22]. However, by nature, the TMDC materials are nonmagnetic because of the absence of magnetic elements and unsaturated bonds. Additionally, the 2H-MoS_2_ phase’s microstructure exhibits nonmagnetic behavior because the spin of two 4d electrons in the Mo^4+^ ions is anti-parallel, resulting in zero magnetic moments. However, the introduction of defects into the pristine TMDC materials produces magnetic behavior [23]. (See Table 1).

## 3. Classification of Defects in the TMDC Materials

The defects that exist in two-dimensional TMDC materials are classified into three categories according to dimensionality as zero-dimensional defects (0D, e.g., point defects, adatoms, and substitutional dopant atoms), one-dimensional defects (1D, e.g., line defects and grain boundaries), and two-dimensional defects (2D, e.g., scrolling, rippling, folding, and heterostacking). The classification of 2D TMDC materials’ defects is shown in Figure 3.

### 3.1. Zero-Dimensional (Point) Defects

In a solid that has an ordered arrangement of atoms in a crystal lattice, any missing atom or the introduction of an unwanted atom at the normal atom position is called point defect. The second law of thermodynamics demonstrates that every solid material has defects [51]. In chemical synthesis, the energy required to change chemical bonds is recovered via the introduction of disorder formation in the crystal. This energy is called “Gibbs free energy”, and it is reversible. The “Gibbs free energy”, denoted by *G*, of a crystal is calculated by the following equation [40]:(1)G=H−TS
where *H*, *S*, and *T* are enthalpy, entropy, and temperature of the crystal. Notably, with the formation of more and more defects, Gibb’s energy decreases continuously, and, at a particular time, it exceeds the minimal energy required for crystal formation, and ultimately no crystal is formed [40]. However, point defects are common in chemical synthesis of 2D materials. The imperfection of chemical synthesis or growing process is the main reason for the point defects. In TMDC materials, six types of intrinsic point defects are observed, including “single sulfur vacancies $\left(V_{s}\right)$, disulfur vacancies ($V_{S2})$, a vacancy complex of a Mo and bonded three S atoms with it on one plane $\left(V_{MoS3}\right)$, a vacancy complex of a Mo and its nearest three disulfur pairs $\left(V_{MoS6}\right)$, and an antisite deformation where a Mo atom was substituting a pair of S (S2) atoms $\left({Mo}_{S2}\right)$ or S2 column occupying the position of a Mo atom $\left({S2}_{Mo}\right)$” [52]. Two types of anti-site defects happen in TMDC materials. One category of anti-site defect occurs via transition metal atoms substituting chalcogen atoms such as ${Mo}_{S}$, ${Mo}_{S2}$, and ${Mo2}_{S2}$. Another type of anti-site defect forms by occupying transition metal atomic places with chalcogen atoms, e.g., $S_{Mo}$ and ${S2}_{Mo}$. However, most of the structural defects in TMDC, ${Mo}_{S2}$, and ${S2}_{Mo}$ anti-site defects are frequently found [9]. The annular dark field (ADF) images of scanning transmission electron microscopy (STEM) shown in Figure 4a demonstrate six types of defects occurring in 2D TMDC materials.

The theoretical calculation of six-point defects, using density functional theory (DFT), based on the energy relaxation method, matches excellently with the experimental method depicted in Figure 4b. Generally, point defects in TMDC materials maintain 3-fold symmetry because the same patterns are obtained three times in $360^{0}$ angular rotation. In contrast, there are some exceptions in TMDC material. For instance, in MoS_2_, an anti-site deformation, ${Mo}_{S2}$, breaks the 3-fold symmetry with one Mo getting closer to two of the three nearest neighbor Mo atoms as well as one S atom between two sulfur layers [52]. Notably, the intrinsic structural stability of point defects depends on the transition metal, such as molybdenum (Mo), chemical potential $\mu_{Mo}$, and chalcogen elements, such as sulfur (S), and chemical potential $\mu_{S}$. Additionally, the chemical potential of TMDC materials is a function of both transition metal chemical potential and chalcogen group chemical potential. In a particular thermal equilibrium, $\mu_{MoS2}=\mu_{Mo}+2\mu_{S}$, where both chemical potentials $\mu_{Mo}$ and $\mu_{S}$ must be less than their values in the bulk counterpart [54]. It is worth noting that the formation energy of mono-sulfur vacancy $\left(V_{s}\right)$ is the lowest among other defects. On the other hand, anti-site ${S2}_{Mo}$ and ${Mo}_{S2}$ formation energies are the highest among other defect formation energies under Mo- and S-saturated environments [52]. Notably, the TMDC materials exhibit contrasting vacancy formation energy with graphene. The divacancy formation energy is lower than monovacancy in graphene [55]. As a result, the mono-sulfur vacancies frequently occur in TMDC materials, where anti-site formations happen irregularly. Interestingly, in TMDC materials, single transition metal vacancies occur in significant amounts, which is volatile [9]. For instance, while MoS_2_ formation occurs once point defects $V_{Mo}$ are produced, the surrounding S atoms are occupying its place by losing 1.1 eV per $V_{s}$ under a S-saturated environment. As a result, in structural defects, a single Mo vacancy is not frequently observed. Most of the Mo vacancies are found in the $V_{MoS3}$ defect complexes. Also, the formation energy of $V_{MoS3}$ is similar to $V_{s}$. However, $V_{MoS6}$ defect complexes are rarely found because the defect formation energy of $V_{MoS6}$ is higher than that of $V_{MoS3}$. Also, $V_{MoS2}$ complexes are not found in structural defects because of their higher formation energy than $V_{MoS3}$ or $V_{S}$ [52]. In TMDC materials, an extrinsic defect is divided into two categories based on the foreign atom’s location to the TMDC crystal lattice. The first one is a substitutional defect that occurs due to the replacement of chalcogen or transitional metal atoms by foreign atoms. The second one is adatoms, which happen when foreign atoms occupy the crystal place of 2D TMDC materials or become trapped in the structural vacancies (M or X position) [53]. The substitution of foreign atoms onto the crystal lattice should follow two basic requirements for the sustained 2D material configuration. The size of the foreign atoms matches the size of the replaced atoms in the crystal. The second one is a post-incorporating lattice structure by substitutional atoms that matches with the original TMDC lattices. As a result, in the periodic table, the elements close to the transition metals or chalcogen elements are potential candidates as the substitutional atoms. Also, lanthanide contraction makes several transition metals feasible as dopants from the transition element series because of their similar radii with 4d and 5d transition metals [56]. Additionally, transition metals and chalcogen elements can be replaced by the same group with similar radius elements as well as similar valences and coordination numbers of the periodic table. For instance, in TMDC crystal, Mo/W or S/Se alloys such as ${Mo}_{1-x}W_{x}S_{2}$ [57], $WS_{2(1-x)}{Se}_{2x}$ [58], and $MoS_{2(1-x)}{Se}_{2x}$ [59] are obtained by CVD, ALD, and PVD methods. Moreover, the Mo/W or S/Se alloy formation depends on the kinetic energy of reaction, ambient pressure, and temperature [2]. The heterostructure alloy could be in or out of the plane. In the in-plane heterostructure, CVD-grown WS_2_ or MoS_2_ triangular domain edges have unsaturated dangling bonds acting as an active growth center that facilitates the incorporation of chemical vapor precursor atoms such as WSe_2_ or MoSe_2_ to extend in the lateral direction of 2D crystals [60]. However, the out-of-plane heterostructure is grown by alternatively keeping one layer of atoms on other layers of atoms [61]. However, as the substitutional atoms, several transition metals except for W with different occupancy of d-orbital than the host atom have been experimentally verified. Manganese (Mn) and rhenium (Re) atoms are both in group 7 of the periodic table, with one extra electron than Mo or W. When Mn or Re atoms are incorporated with MoS_2_ or WS_2_, they become n-type material. Both Mn and Re could be stable substitutional atoms if the doping concentrations in the MoS_2_ or WS_2_ are 2% and 0.6%, respectively. Otherwise, the 2D nature of MoS_2_ or WS_2_ cannot be preserved. Also, dopant atoms will remain separated from the pristine crystal [2]. Another substitutional atom is niobium (Nb), which acts as a p-type dopant because of its one less electron than the host atom in the MoS_2_ or WS_2_. The 2D structural morphology of TMDC will remain preserved if Nb’s doping concentration does not exceed 6.7% [2]. Other substitutional atoms are Cr, V [62], and Au [63]. Likewise, using density functional theory (DFT), some computational works have been performed to substitute chalcogen atoms from TMDC. One of the potential groups in the periodic table is the halogen family, group 17, such as F, Cl, Br, and I. These substitutional atoms act as n-type doping for MoS_2_ since halogen group elements have one excessive p electron with respect to S. On the other hand, p-type substituting elements for the chalcogen site in TMDC are the group V elements, including N, P, and As in the periodic table [64]. The structural substitutional dopant atoms are identified by the variation in “Z-contrast” in ADF-STEM images shown in Figure 5a. Additionally, to prevent misidentification of the substitutional defects at the transition metals, sites are confirmed jointly by using ADF-STEM and atomic resolution of EELS (“Electron energy-loss spectroscopy”) [65]. The rotational defect is another type of zero-dimensional defect in the TMDC materials. This defect is found only in WS_2_ and MoSe_2_ TMDC materials. In this rotational symmetry, the bonds of three chalcogen atoms with the same metal atom rotate 60° and cause defects while keeping the original lattice symmetry. These trefoil defects are found abundantly at elevated temperature. The vacancy transformation depends not only on bond rotation but also on migration and reordering of the chalcogen atoms to the high-concentration vacancy region. The mechanism of rotational defect is represented in Figure 5b [66]. Additionally, the foreign atoms not only substitute host atoms of the lattice but can also be absorbed on the lattice plane. When the atomic radius of foreign atoms is larger than host atoms, they occupy the TMDC surface instead of the lattice point. Lattice strain and distortions are generated due to these adatoms. Depending on the adatoms adsorption sites on the TMDC planes, the functionalities and nanoscale applications are changed. In the 1H-phase TMDC materials, there are four possible places for adatoms to be adsorbed before structural optimization. The possible sites are top of the transition metal atom $\left(T_{M}\right)$, top of the chalcogen atom $\left(T_{Ch}\right)$, between or on the transition metal–chalcogen atomic bonds (B), and void of the hexagonal plane $\left(C\right)$, as shown in Figure 5c [54]. However, in 2H-TMDC, there are two places on which adatoms are adsorbed, including the hexagon’s void center and between the atomic bond of transitional metal and chalcogen atoms [67]. 

In alkali atoms, the transition metal site is the most energetically stable site for these electron-donating adatoms on the TMDC materials. There are several adatoms on TMDC materials, including Pt, Co, and Au. In the ADF-STEM analysis, it is found that Pt adatoms locate on the S sites. The Pt adatoms have high mobility on MoS_2_ because of the lower migration barrier in MoS_2_. However, Co atoms bind not only above Mo atom but also on the hexagonal hollow. On the other hand, Au adatoms are bound either above S or above Mo, or void site of the hexagonal. It is worth noting that most of the adatoms observed on the MoS_2_ are Mo or S. High-angle ADF (HAADF) analysis shows that adatom S is mostly stable at $T_{Ch}$ site, but Mo adatoms are stable both at $T_{M}$ and C [54].

### 3.2. One-Dimensional (Linear) Defects

Line vacancy of TMDC films is caused due to the agglomeration of chalcogen atoms vacancy on the film surface. Single and double S vacancies occur with the zigzag orientation confirmed by high resolution transmission electron microscopy (HRTEM) exhibited in Figure 6a because this orientation possesses the lowest energy [2]. The result of theoretical calculation matches with the experimental work shown in Figure 6b [2]. The line vacancy depends on the number of S vacancies and the strain on the surface that selects the line vacancy ordering. Grain boundaries (GBs) in 2D transitional metal dichalcogenides are formed by the stitching of atoms of two nearest lattice mismatch grains. However, the binding nature of transitional metal atoms and chalcogen atoms forms various dislocation cores. In MoS_2_, the bonding behavior between Mo and S renders conventional 5|7 folded rings and forms 4|4, 4|6, 4|8, and 6|8 folded dislocation cores. When these dislocation cores attach, it forms grain boundaries [69]. As TMDC monolayer composed of three atomic layers, such as chalcogen–transition and metal–chalcogen, the grain boundaries and dislocation core formation are quite difficult. Moreover, if the atoms are dislocated from the TMDC structure, the lattice structures are reoriented and form various ring-shaped dislocations depending on the grain boundary’s angle [69]. The parallel grain boundary on the zigzag orientation of TMDC lattice is composed of four-sided rings with two shared points, denoted as 4|4P, exhibited in Figure 6c, at a common chalcogen (S or Se) site in MoS_2_ or MoSe_2_ [2,70].

This point shared with two nearest chalcogen atoms forms mirror twin grain boundaries when grains make a 60° angle towards zigzag orientation. In this grain boundary, Mo atoms maintain a 6-fold structure, but S or Se atoms change coordination from 3-fold to 4-fold [52]. In this parallel grain boundary, 4|4P folded rings are connected by octagonal kicks that sometimes absorb chalcogen atoms inside them because of sufficient space, making octagonal kicks into two distorted hexagonal [52]. Another type of 60° grain boundary in TMDC materials is formed when two grains are moved half of the primitive lattice vector to the grain boundary. Actually, in this grain boundary, the two four-fold rings share edges at the metallic sides denoted as 4|4E, shown in Figure 6d [52]. Indeed, low angular edges make a complicated situation mainly in the 4|6-, 4|8-, 5|7-, and 6|8-sided grain rings because these low angular grains experience high strain up to 58% and chalcogen atoms become mobile even after the low amount of accelerating voltage, producing grain dislocations in the lattice. This is confirmed from the ADF-STEM images of WS_2_ monolayer films. These images confirm that, in the monolayer WS_2_ films, the movement of grain’s boundaries occurred due to the migration of atomic dislocation in the monolayer WS_2_ films [73]. Moreover, one of the critical defects in the TMDC materials is their edge disorders. In synthetic TMDC materials, the most-adopted single-crystalline flakes are triangular. According to Wulff construction, in 2D materials, triangular edges exhibit low energy shape; for that reason, triangular edges are frequently found in TMDC materials’ deposited films [2]. The surface energy of TMDC materials’ flakes depends on the sulfur chemical potential $\mu_{S}$ or the sulfur vapor potential. When the pressure of sulfur vapor is low, unstable hexagonal distorted shapes are formed instead of stable triangular shape. The reason behind this scenario is that, during low-pressure sulfur vapor, the chemical potential of sulfur is low. The theoretical nanoscale calculation using DFT showed that, in a S-rich environment, the most thermodynamically stable condition for MoS_2_ film is the Mo-terminated edges with 50% or 100% S coverage [70,74]. While it is demonstrated that, in CVD-grown MoS_2_ films, the Mo-terminated edges with both 0% and 50% S coverage are observed, CVD-grown TMDC materials films are far away from thermodynamically stable condition [52]. In CVD, a sharp gradient of MoO_3_ precursor on the growth substrate impacts the MoS_2_ domain growth at a uniform temperature. The shape change in the crystal domains is due to the local concentration change in the Mo: S ratio, and this concentration change influences the dynamic growth of edges [72]. In this method, some precursors, including (NH_4_)_2_MoS_4_, MoO_3_, MoCl_5_, and Mo film, have been employed to synthesize various shapes, such as triangle, three-point star, and hexagons [75]. The shape evaluation depends on the growth kinetics, ambient temperature, pressure of the inert gas flow, and the distance between transitional metals and chalcogen precursors in the reaction chamber. According to the crystal growth mechanism, the slow-growing faces become the largest, while the faces that are sharply growing either disappear entirely or shrink [76]. In 2D materials, the growing rate of faces depends on the edge of free energy. The final shape of monolayer MoS_2_ depends on the growing rate of different edge terminations, as explained in Figure 6e. The most energetically stable and commonly observed edge structures are “Mo zig-zag (Mo-zz) terminations and S zig-zag (S-zz) terminations” [75]. Also, in the S-zigzag-terminated edges, each single S atom is bonded with nearest two Mo atoms, and these chalcogen atoms remain outside, whereas, in the Mo-terminated zigzag edges, each exposed Mo atom is bonded with four neighbor S atoms. This structural difference has a great impact on the shape evaluation because each structure’s chemical activity becomes different from the variation in the Mo: S ratio. The three conditions of Mo:S ratios (>1:2, 1:2, <1:2) dominate the various shape formations. In the Mo-abundant atmosphere in the reaction chamber, when Mo: S ratio is greater than 1:2, the S-zz-terminated edges are more energetically unstable than Mo-terminated edges because, in these circumstances, S-terminated edges grow faster than Mo-zz. Moreover, there is a higher probability of exposed unsaturated S atoms forming bonds with free Mo atoms. As a result, the domain shapes are changed from hexagonal to triangular with three Mo-zz-terminated edges. On the other hand, when the atomic ratio of Mo and S in the reaction ambient is same as the chemical stoichiometric ratio of MoS_2_ and the ratio of Mo:S equal to 1:2, then the two types of edge terminations, probability, and stability, occur corresponding to both Mo and S free atoms. In this case, the hexagonal domain shape is obtained at the final stage. Moreover, in the S-enhanced environment when Mo:S < 1:2, unstable Mo-terminated edges grow faster than S-terminated edges. Therefore, there is a higher probability of exposed Mo atoms meeting and bonding with saturated S atoms. In these circumstances, the final shape is converted from hexagonal to triangular with sulfur-terminated edges [72]. Additionally, the exact ratio of Mo and S on the substrate surface influences the shape formation. For instance, if Mo and S ratio is slightly greater than 1:2, the growth rate and termination of Mo-zz and S-zz are nearly equal, and the truncated shape is grown at the same time interval subjected to other stated conditions. The shape change depends on the horizontal pulsing gas flow direction and depends on the vertical direction from the substrate. Moreover, the precursor temperature has a significant influence on the shape formation of TMDC films. For instance, if the MoO_3_ precursor’s temperature is altered from 700 °C to 650 °C, three-point stars domain shape is formed instead of a triangular shape. The reason for shape changing is that, due to the precursor temperature decline, a decrease in Mo concentration results. As a result, an edge termination difference is produced between Mo-zz and S-zz and the resulting shape is changed [72].

### 3.3. Two-Dimensional (Planar) Defects

Both practical and experimental evidence show that perfect atomically thin two-dimensional crystals cannot exist thermodynamically. Any finite thermal fluctuations break the periodicity of the 2D lattice, and, finally, it is melted. However, after thorough practical investigation on 2D graphene, it is evident that two-dimensional materials’ durability depends on either their specific size or their crystal imperfections [77]. In TMDC materials, localized strain forms ripple deformation shown in Figure 7a that greatly impacts these materials’ physical properties. The local strain is not uniform and exists in the TMDC nanostructures at the step edges during film formation because of the lattice mismatch with the substrate. For example, when the vapor-phase-deposited MoS_2_ nanostructures are cooled down at room temperature, ripples emerge on the film surface because of the local strain relaxation due to the thermal expansion effect. Intrinsic strain due to lattice mismatch and thermal expansion difference between substrate and nanostructure of TMDC materials significantly influence the height and wavelength of the ripple deformations [78]. 

The uniaxial local strain ε on top of the ripple MoS_2_ film is calculated from the following equation [79]:(2)$$\varepsilon =\frac{\pi^{2}nh}{\left(1-\sigma^{2}\right)\lambda^{2}}$$
where *n* is the thickness of MoS_2_ nanostructure, σ is the Poisson ratio, *h* is the height, and λ is the ripple’s wavelength. In monolayer rippled MoS_2_ film, the range of strain is from 0.001 to 0.006 for different ripple heights and wavelengths. The investigation of surface potential and charge distribution using Kelvin Probe Force Microscopy (KPFM) on monolayer MoS_2_ film reveals that the charge distribution is inhomogeneous due to the ripple deformation. Moreover, this local strain changes bond lengths, bond angles, and corresponding surface curvature of MoS_2_ that shifts the energy level toward the Fermi level. Conversely, the surface potential and charge distribution on atomically thin MoS_2_ film show homogeneity and are distributed uniformly, respectively [78]. Also, the average ripple height of TMDC materials is nanometer in range, and the ripples are formed deliberately using laser scanning on the film surface. The ripples on the TMDC films’ surface tune the optoelectronic properties of TMDC materials. In contrast, in the multilayer TMDC materials, each layer is piled on top of one another by van der Waals (vdW) forces. These forces depend on the nearest interlayers’ spacing. Because the interlayer spacing is varied according to the atomic stacking of various phases, the phases of TMDC materials follow the Bernal sequence (such as ABA, ABC, etc.). However, when the flakes of TMDC materials are stacked one after another by the synthesis process, then deviation regarding Bernal stacking of these materials could happen. Also, during exfoliation and transfer of flakes, 1H phase sheets can be folded on themselves. Since the vdW interaction has an impact on the various properties of TMDC materials, the atomic stacking of phases and layer orientation can be a 2D defect depicted in Figure 7c [2]. Moreover, the 2D materials, e.g., “TMDCs”, “h-BN”, and “graphene”, are vertically stacked on top of one another. In this 2D materials heterostructure, with the stacking of adjacent layers represented in Figure 7b, a similar crystal structure is formed by the vdW epitaxy that tolerates the lattice mismatch of the adjacent layers. Due to the lattice mismatch, these heterostructures generate 2D defects, resulting in new properties of the materials. (See Table 2).

For instance, the heterostructures such as MoSe_2_/MoS_2_ and graphene/WSe_2_ formed by van der Waals (vdW) epitaxy generate regular moiré patterns indicated in Figure 7c. Each moiré pattern creates the individual optical properties of the heterolayer materials. However, the other vdW hetero-epitaxy properties are not well-known; thus, more works, both theoretical and experimental, are needed to understand the diverse nature of these heterostructures [2].

**Figure 7 nanomaterials-14-00410-f007:**
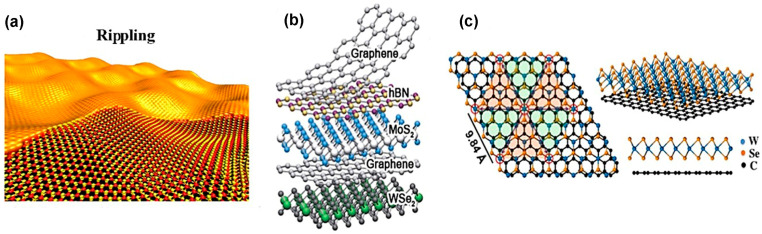
Two-dimensional defects: (**a**) rippling. Reprinted with permission from Ref. [2]. 2016, IOPscience (**b**) heterostructure of various 2D materials Reprinted with permission from Ref. [87]. 2013, Nature; and (**c**) moiré pattern of vertically stacked monolayer WSe_2_-graphene. Reprinted with permission from Ref. [88]. 2015, American Chemical Society.

## 4. Defect Characterization Tools for 2D TMDC Materials

Layered two-dimensional transition metal dichalcogenide and artificial heterostructures consisting of different 2D materials have gained attention in the research community because of their promising optical and electronic properties. However, the identification of the exact deposited layers or their transformation to the heterostructures, quantifying defects on the individual layers or within the layers while stacking the variety of 2D layers, and the doping levels are significant entities during the fabrication of 2D-materials-based devices. For structural analysis, Raman spectroscopy, transmission electron microscopy (TEM), scanning tunneling microscopy (STM), scanning electron microscopy (SEM), and atomic force microscopy (AFM) have been discussed for the 2D TMDC materials.

### 4.1. Raman Spectroscopy

Raman spectroscopy is a non-destructive spectroscopy technique used to identify molecular vibration, rotation, and other states of particular materials. The surface morphology, information of chemical fingerprint, the lattice structure of edges, and even electronic properties of atomically thin 2D materials are obtained from this analytical spectroscopy [89]. The theoretical analysis predicted the four Raman active modes, i.e.,$E_{1g}$, $E_{2g}^{1}$, $E_{2g}^{2}$, and $A_{1g}$, and two inactive modes, such as $B_{1u}$ and $B_{2g}$, towards c-axis of transition metal dichalcogenides, as shown in Figure 8a [89]. Among four active modes, the $E_{1g}$, $E_{2g}^{1}$, and $E_{2g}^{2}$ are in-plane modes and $A_{1g}$ is out-of-plane mode. Indeed, the two active modes are generally identified in the 2D TMDC materials. One is symmetric in-plane lattice vibration mode $E_{2g}^{1}$ and the other is out-of-plane lattice vibration mode $A_{1g}$. Because the vibration frequency of $E_{2g}^{2}$ mode is very low, ~30 cm^−1^, which is not accessible, and due to the backscattering on the basal plane, $E_{1g}$ mode is forbidden [89]. However, the number of layers of 2D TMDC materials is determined using Raman spectroscopy. The line shape of the 2D band depends on the stacking order, layer number, and the excitation energy of the laser power in Raman analysis [90]. In 2D TMDC materials, the choosing of excitation energy in Raman measurement is an important criterion because the optical interference on the substrate of these materials greatly affects the Raman intensity [91]. Additionally, the minimum wavelength of the laser depends on the thickness of both the sample and SiO_2_ on the Si substrate [90]. In MoS_2_, using appropriate excitation energy, the separation between two active modes $,E_{2g}^{1}$ and $A_{1g},$ is used to determine the layers’ number in the sample demonstrated in Figure 8b [90]. Because the polarization and symmetry of vibration of these two modes are different from each other, however, the two active modes strongly depend on the layer number. They can be selectively switched between on or off state by polarization [92]. However, in TMDC materials including MoS_2_, MoSe_2_, MoTe_2_, and WS_2_ [93], the $A_{1g}$ mode is shifted towards the red end with the decreasing number of layers because the lowering of restoring force is affected in the layered structure, whereas the $E_{2g}^{1}$ mode is shifted to the blue end with the reduction in the layer number due to the interlayer interactions or intralayer bonding [89]. Contrastingly, the Raman spectrum of WSe_2_ follows the layers’ dependence trend, but degeneracy occurs in monolayer or few layers of this material [94]. The shifting of Raman peaks decreases with the increasing weight of MX_2_ molecules [93]. The layer-dependent Raman mode frequency is also found in the unconventional 2D materials, including 1T’-PtS_2_ and phosphorene [93]. On the other hand, the Raman spectra of 1T-HfS_2_, 1T’-ReS_2_, or 4H-SnS_2_ do not depend on the layers because of the strong electronic coupling layers [95]. However, there are some limitations to identify the number of layers using active modes of Raman spectroscopy. 

Second, these vibration modes depend on the lattice strain, doping type and doping concentration, and the substrate, so there is a high chance of incorrect identification of layer numbers. The alternative way to identify layer numbers more accurately is by comparing the breathing and interlayer sharing modes shown in Figure 8b. The reason behind the accuracy of this technique is that the shifting of the modes depends on the layer number, and the vibrations are rigid along both in-plane and out-of-plane directions [90]. Additionally, the laser power of Raman spectroscopy is used to study the impact of temperature on the properties of TMDC materials. Generally, the Raman peaks of the TMDC materials are shifted towards the red end and can also give rise to additional secondary phases with increasing laser power. For example, in single-layer MoS_2_, the red shifting of the two active modes, such as $E_{2g}^{1}$ and $A_{1g}$ peaks, occurs with increasing the laser power from 0.04 to 0.164 mW [97]. Moreover, Yan et al. showed that the variation in parameters of the two modes occurred non-linearly when the laser power was greater than 0.25 mW [97]. The redshifting of two peaks of the active mode also occurs in other 2D materials, including WS_2_ [98] and BP [99]. Moreover, the electronic band dispersion and mechanical properties of the ultra-thin TMDC materials are highly sensitive to external strain. Raman spectroscopy is an excellent tool to investigate the applied strain to the monolayer and a few layers of TMDC materials. Indeed, the applied strain above the critical value on the ultra-thin TMDC materials breaks the crystal symmetry; thus, the in-plane mode $E_{2g}^{1}$ breaks into two modes [100]. For instance, Wang et al. demonstrated that, after increasing the applied strain on monolayer MoS_2_, the active mode $E_{2g}^{1}$ splits into two modes under ~1% uniaxial tensile strain [101]. While studying the effect of substrates, the Raman in-plane vibrational mode $E_{2g}^{1}$ is insensitive to the substrate, although the position of out-of-plane mode $A_{1g}$ varies towards the blue end. The shifting of $A_{1g}$ mode towards the blue end occurs due to the weak vdW interaction between the 2D materials and the substrate. For instance, the frequency shifting of out-of-plane mode ($A_{1g}$) of MoS_2_ was varied on SiO_2_ and quartz substrate [102]. Banszerus et al. demonstrated that h-BN and graphene are better substrates for 2D TMDC materials [103]. Additionally, the rough and oxidizing substrate surface is not suitable for Raman analysis because these surfaces interfere with the Raman active modes [100]. The out-of-plane mode $A_{1g}$ is sensitive to doping. Chakraborty et al. reported that the redshift of $A_{1g}$ mode is observed in MoS_2_ as gate oxide after applying gate voltage due to strong electron–phonon coupling. However, in-plane mode $E_{2g}^{1}$ is insensitive to doping [104]. Furthermore, the defect density is estimated using Raman spectroscopy. The frequency shifting and linewidth evaluation occurs with increasing defect density. A Raman peak at ~220 cm^−1^ is observed due to the longitudinal acoustic (LA) phono vibration. This LA mode is proportional to the defect density. Therefore, the intensity of LA (M) mode is used to determine the defect density in 2D TMDC materials [96]. The tip-enhanced Raman spectroscopy (TERS) signal is used to identify the edge structure of TMDC materials by the plasmonic Au or Ag tip demonstrated in Figure 8c [96]. The investigation of the periodic arrangement of different atoms of TMDC materials, either zigzag or armchair, is important because they impact electronic, optical, and magnetic properties of these materials. In graphene, the edge structure is determined using the D-peak because it is active at the armchair edges and inactive at the zigzag edges. In TMDC materials, the peak of out-of-plane mode $A_{1g}$ was found in the opposite direction at 406 cm^−1^ wavenumbers in Figure 8d, when the tip was moved toward the edges from the basal plane. The downshift peak was observed in zigzag MoS_2_ edges in Figure 8e, and the upshift peak was observed in armchair MoS_2_ edges in Figure 3 [96].

### 4.2. Transmission Electron Microscopy (TEM)

In transmission electron microscopy, the interaction of transmitted electrons from the sample builds the image [105]. The TEM imaging electrons interact with the sample elastically or inelastically; thus, the specimen faces charging, heating, atomic displacement, and ionization problems [106]. However, this specimen disorder can be minimized by choosing the appropriate acceleration voltage as long as it does not cross the threshold energy; above the threshold energy, atomic displacement occurs. However, in the in-situ TEM, the unraveled atom-by-atom dynamic process facilitates investigation of phase distortion, atomic dislocation, and grain boundaries. Moreover, Dumcenco et al. showed that it is possible to ensure the chemical composition of the investigated materials using STEM. Scanning transmission electron microscopy (STEM) can track the specific atoms on the surface because the imaging in STEM is related to the specific atomic weight (Z-contrast) on the surface [105,107]. The surface structures with atomic resolution of the 2D TMDC samples or their heterostructures are investigated using transmission electron microscopy (TEM) at the minimized acceleration voltage even during aberration correction. However, the transferring process of the prepared 2D TMDC materials’ film on the TEM grid from the substrate is an important factor because of the challenges of avoiding contamination and sample degradation. Generally, the 2D TMDC materials’ flakes are transferred on the TEM grid using two methods, including the wet chemical process and ultrasonic bubbling process [105]. In the wet chemical process, the prepared TMDC flakes on the substrate are coated with poly methyl methacrylate (PMAA). After that, the substrate is etched with KOH chemically, and the TMDC flake with a PMAA supporting layer is removed, as shown in Figure 9a. Then, the TMDC film is placed on the TEM grid, and the PMAA is removed in the liquid or vapor form by dissolving in the solvent [108]. On the other hand, in the physical exfoliation method, the deposited TMDC films on the PMAA-coated substrate are submerged in water, and a micron-size bubble is produced in the water using ultrasonication. During this sonication process, the micron-size bubbles interact and make the layers collapse; thus, they generate force at the interface between the substrate and the TMDC flake. This acting force is high enough to detach the TMDC-PMAA layer from the substrate, as depicted in Figure 9b [109].

After detaching from the substrate surface and placing on the TEM grid, the exfoliated TMDC flake is investigated using numerous TEM techniques. In conventional TEM imaging, the structural information of TMDC materials is obtained. On the other hand, high-resolution STEM imaging can provide structural and chemical information from Z-contrast. For further confirmation of chemical composition, the selected areas of the TMDC materials are investigated using EELS (electron energy loss spectroscopy) and EDX mapping. This mapping provides more detailed information about atomic distribution, heterojunction, and number of layers [105]. However, point defects, dislocations, and grain boundaries of the TMDC materials can be observed using annular dark field (ADF) imaging of STEM. Zhou et al. reported six types of point defects of CVD-grown MoS_2_ films using ADF-STEM imaging [96]. Also, Lin et al. demonstrated the rotational defects formed due to the Se vacancies in WSe_2_ monolayer using ADF-STEM imaging [66]. Moreover, Liu et al. used time-resolved annular dark field imaging to identify single point defects in a monolayer WS_2_ film [28]. The formation and structure of dislocations into the grain boundaries depend on the type of dislocation and the constituent parts in the TMDC materials. Also, during the synthesis of two-dimensional TMDC materials, the randomly oriented grain boundaries change these materials’ electronic and optical properties [110]. Therefore, atomic investigation is required to control the atomic structure of the grain boundaries of TMDC materials because it is an important factor in tailoring the various properties of these materials. Zhou et al. investigated the formation of antiphase boundaries on monolayer and bilayer MoS_2_ films using ADF-STEM imaging. These antiphase boundaries affect the electronic properties of the MoS_2_ [111]. Ly et al. demonstrated the atomic structure of MoS_2_ grain boundaries using high-resolution TEM by transporting electrons through single-grain boundaries of this material [112]. Also, using ADF-STEM, van der Zande et al. reported the zigzag and twin grain boundaries of MoS_2_ [75]. Garcia et al. investigated the electron-irradiated defect formation on MoS_2_ films [113]. Aside from that, Lin et al. demonstrated the phase transition of MoS_2_ from 2H to 1T using ADF-STEM imaging [74]. Additionally, the lateral and vertical hetero-microstructure of the hybrid TMDC materials or different 2D materials can be characterized by high-resolution transmission electron microscopy and selected area electron diffraction (SAED) pattern. High-resolution transmission electron microscopy can observe the lateral heterostructures between different TMDCs or TMDC and other 2D materials. Fu et al. reported the heterostructure formation of MoS_2_ and h-BN layers by revealing different layers using STEM [114]. Also, the vertically layered structure of hybrid TMDC materials and other 2D materials can be observed from the cross-sectional view of the TEM images. Lin et al. reported the apparent vertical stacking of monolayer MoS_2_ and WS_2_ on top of a few layers of graphene by high-resolution transmission electron microscopy. They also found that the graphene surface defects lead to multilayer growth of TMDC materials [115]. Moreover, TEM can investigate the electronic properties of the TMDC heterostructures by measuring the bandgap of the total heterostructure. For example, Hill et al. reported that the MoS_2_/WS_2_ heterostructure bandgap was 1.45±0.06 eV using the spectroscopic investigation of the TEM [116]. Also, using the electron energy loss spectroscopy (EELS) in TEM, it is possible to measure the bandgap of the TMDC materials’ heterostructures by filtering the incident electron beam to a fraction of the energy distribution [105].

### 4.3. Scanning Tunneling Microscopy (STM)

Scanning tunneling microscopy (STM) works based on the scanning probe microscopy (SPM) technique for investigating the 2D TMDC materials. During scanning, the position of the conducting atomically sharp STM tip has to be maintained in proximity with the sample surface so as to cause tunneling current. The feedback bias voltage controls the vertical position of the tip. This feedback voltage allows the electrons to tunnel through the gap between the tip and the sample surface. The tunneling current variation depends on the sample’s local density of states, the bias voltage, and the tip position [105]. The STM is used for atomic resolution imaging to investigate the sample’s electronic states, and to perform atomic-scale nanomanipulations with the clean and stable surface in the vibration-protective environment [105]. However, the energy of the tunneling current in the STM is low; therefore, no surface modification or disruption of the intrinsic defects of the TMDC materials happen; thus, the distribution of the defect density is measured directly using STM [117]. Although STM is employed to investigate the native defects of the TMDC materials surface, it is challenging to resolve the image with atomic-scale point defects in monolayer or few-layered TMDC materials [118]. Additionally, the previous STM investigations demonstrated the modification of defect density states at the nanometer scale, but no imaging was achieved on the atomic scale. This could be due to the fact that the native defect states lie within the wide bandgap of the TMDC materials. Therefore, the STM imaging leads to inaccurate defect density in the TMDC materials [117]. Also, it is difficult to obtain dynamic information about the defect densities with the TMDC specimens. For instance, Vancso et al. demonstrated the atomic-scale imaging of point defects in monolayer MoS_2_ surface areas. They achieved discontinuous circular or triangular point defects on the MoS_2_ surface using STM, as shown in Figure 10a,b [117]. 

Also, nanomanipulations, such as the cutting down of monolayer TMDC films or the rotation of this film in any direction on the substrate, can be achieved using STM [105]. For example, Koós et al. reported the cutting down monolayer MoS_2_ on the graphite surface using STM lithography. They created MoS_2_ nanoribbon by cutting it down from 18 nm to 12 nm in a variety of specific crystallographic orientations, as represented in Figure 10c–f [119]. Also, they could rotate or move these nanoribbons in many intended directions on the graphite substrate [119]. These nanomanipulation abilities indicate the possibility of assembling complex and miniaturized devices using STM.

### 4.4. Scanning Electron Microscopy (SEM)

Scanning electron microscopy (SEM) produces the surface image by scanning the sample using a focused electron beam. The interacting electrons with the surface atoms of the sample produce signals that can provide information about the composition and surface topography of the sample. As the electron beam from the SEM excites the surface atoms of the sample, the atomic excitations release secondary electrons detected by the secondary electrons’ detector of the SEM [105]. The intensity of the produced signal depends on the number of secondary electrons detected. However, no atomic resolution imaging has been reported using SEM. However, SEM can characterize low-resolution and large areas of the sample that provide morphologies, including the shape and size of deposited flakes of the 2D TMDC materials. For example, Xie et al. used SEM to characterize the surface morphology of the chemical-vapor-deposited monolayer MoS_2_ flakes. They were able to achieve the shape evaluations, including triangular, truncated triangular, and hexagonal-shaped flakes using different concentration ratios of Mo and S, as shown in Figure 11a–c, respectively [120]. Also, the number of layers of deposited TMDC materials and the heterostructures could be identified by combining SEM and EDS. For instance, Lang et al. reported the characterization of MoS_2_ films using SEM and EDS. They identified the number of layers of MoS_2_ on the silicon substrate as well as the thickness of the individual MoS_2_ flakes, as shown in Figure 11d. They were also able to identify the layer number and the thickness of the individual layer of the MoS_2_/WSe_2_ heterostructure [121].

### 4.5. Atomic Force Microscopy (AFM)

Atomic force microscopy (AFM) is a powerful tool to characterize 2D TMDC materials. AFM works based on the scanning probe microscopy (SPM) technique. AFM scans the sample surface using a cantilever with an atomically sharp tip; together, this assembly is called the probe [105]. The AFM probe scans the specimen surface in a raster pattern. When the AFM probe comes in proximity to the specimen surface during scanning, the short-range attractive or repulsive forces between the AFM probe and the sample surface deflect the vertical position of the probe. The vertical and lateral movements of the AFM probe are monitored by a laser beam, which is reflected off the cantilever. The position-sensitive photodetector captures the reflection of the laser rays due to the vertical and lateral movement of the cantilever, and this capturing laser ray is converted to the surface topographic images. However, the contaminations cause discrepancies during AFM imaging. Because the AFM imaging is carried out in the open environment, introduction of contaminant layers, such as hydrocarbon or water vapor, on the sample surface occurs [105]. These contaminant layers degrade the image resolution because the contaminant layers cause large surface areas between the probe and the sample surface and return the false destabilized feedback potential to the AFM. The surface contaminants before AFM investigation can be removed by annealing. For example, Godin et al. removed the water layer from the WS_2_ surface by using high-vacuum annealing. The variation in step height of the AFM images was reduced by surface water removal because it reduces the experienced capillary forces of the tip. It is worth noting that the long-range capillary forces disappeared after annealing, but the short-range, approximately a couple of nanometer ranges, capillary forces remained due to the attractive forces [122], as shown in Figure 12a,b. However, AFM is used to investigate the surface morphology, edge effect, and property changes in TMDC materials. It is possible to know the number of layers from the height profile of the AFM images because the step height at the edges indicates the thickness of the nanosheet. This measurement cannot provide the exact number of layers but indicates rough estimation based on the measured profile height variation over the sample surface. It should be noted that the thickness of the monolayer films varies with the different TMDC materials [123]. Therefore, it is difficult to determine the number of layers of TMDC materials from the AFM height profile if the thickness of the monolayer film of TMDC materials is unknown. The height profile measurement using AFM is shown in Figure 12d [123]. AFM can investigate the edge effect on the formation of wrinkles and buckles of the TMDC materials. Ly et al. reported the edge effect on the buckle formation of WS_2_ film. They investigated as-grown monolayer WS_2_ film with a thickness of 0.8 nm using contact and tapping mode of atomic force microscopy. 

The buckling of the flakes was observed in the tapping mode in the brighter domain of the WS_2_ film (Figure 12e) because a height contrast at the edges of the film was observed in this mode, while the flat surface was observed in the contact mode, as shown in Figure 12f [125]. Therefore, the buckling effect depends on the contrast of the edges of the TMDC materials’ flakes, and it does not depend on the intrinsic defects of these materials [125]. Additionally, the contact between TMDC materials and metal introduces a Schottky barrier, limiting the current injection through this junction. This Schottky barrier height is measured by the conductive mode of atomic force microscopy (C-AFM). In this mode, the Pt-coated tip acts as a nanoscopic metal electrode, as shown in Figure 12c [124]. Giannazzo et al. reported the variation in Schottky barrier height from the recorded I-V characteristics of Figure 12c. Moreover, the conductive mode of atomic force microscopy is also used to investigate the local defects, charge variation, and stacking fault of the TMDC materials [126]. Also, the physical properties changes in TMDC materials due to solvents such as acetone, chloroform, and toluene can be investigated using the conductive mode of AFM. For example, Choi et al. reported the carrier density changing of MoS_2_ and WSe_2_ in chloroform, acetone, and toluene solvents using C-AFM [127]. 

## 5. Defect Engineering

### 5.1. Defect Engineering during Synthesis

In TMDC materials, several types of defects exist. The defect density and the type of defects of as-prepared TMDC materials are different if their synthesis processes are varied. For example, in mechanically exfoliated (ME) and CVD films, vacancies of chalcogen atoms are frequently found. On the other hand, in CVT and PVD, the most dominant defects are transition metal vacancies and anti-site defects [9]. The defect densities of these TMDC materials can be tuned by changing the input parameters related to the synthesis processes adopted. The comparison of various defects’ formation procedures in different synthesis processes is shown in Figure 13a. In mechanically exfoliated films from the TMDC materials either extracted or grown naturally or artificially, chalcogen vacancy is a prominent defect. The reason is that, during deposition of MoS_2_, the higher saturated S atoms are released more than Mo atoms from the gas phase of MoS_2_. Therefore, S deficiency occurs, leading to mono-sulfur vacancy and disulfur vacancy, but no anti-sites are formed. However, controlling sulfur vapor during MoS_2_ formation will minimize S deficiency in the bulk MoS_2_, leading to defect-free or decreased chalcogen atoms vacancy in the mechanically exfoliated samples [9]. In PVD, the solid precursors of MoS_2_ are converted into clusters and atoms in the gaseous phase. These atoms and clusters are transported by inert gases and condensed into solid phase of MoS_2_. However, in gas phase, S has saturated vapor pressure; as a result, during transportation, it leaves the reaction chamber more than Mo atoms, leading to S-deficient environment. The high mobility of clusters and atoms in the S-deficient environment tends to create the lowest energy structure. Due to this Mo-rich environment, not only MoS_2_ but also anti-sites such as ${Mo}_{S}$ or ${Mo}_{S2}$ are formed at a time. The anti-site defects can be minimized during synthesis by controlling the S vapor’s transportation rate or reducing the surplus of Mo atoms [9]. In a typical CVD process, during the synthesis of TMDC materials, the transition metals from its precursors are chemically reduced by chalcogen atoms. Due to the shorter synthesis time, low deposition temperature, and volatile precursors, the engineering of defects in TMDC materials is feasible in this process. For instance, the edge formation and density of S vacancies of MoS_2_ can be tuned by controlling the sulfur vapor in a lateral direction [9]. Additionally, it is feasible to maintain the shape and size formation of TMDC materials such as “isolated WS_2_ triangular domains” and “quasi-continuous WS_2_ films” by controlling the reaction chamber pressure. As a result, the grain density is controlled by maintaining the grain’s average size and shape. Moreover, the heterostructure interface could be formed in either 1D or 2D by controlling the growth temperature in the CVD reaction chamber [2]. Additionally, the properties of as-prepared TMDC materials can be altered by a two-step process of CVD. These two steps are performed when the precursors and deposition conditions are different from each other, and this two-step process is used during TMDC alloy formation. For example, the monolayer of $MoS_{2x}{Se}_{2(1-x)}$ is formed by either selenization of MoS_2_ or sulfurization of MoSe_2_, and this monolayer alloy exhibits tunable bandgap [81]. The CVD-grown monolayer alloy suppresses the Se vacancy by approximately 50%. Thus, anti-site defect control is obtained in the CVD method. Some residual O atoms create bonds with Mo atoms, taking S atoms’ position in the CVD process. However, due to O atoms’ low stability, it is desorbed into the gas and creates a vacancy in the S site. It is assumed that Mo occupies this place and creates anti-sites [9]. Indeed, CVT is a comparatively better method to synthesize large stoichiometric TMDC crystal or monolayers or multilayers of TMDC films. In this method, the TMDC powder is sealed in the quartz tube, and this process requires high growth temperature and long synthesis time. Various transport agents are used, including I_2_, Br_2_, TeCl_4_, and H_2_O vapor [128]. The as-grown TMDC films show uniform crystallinity and free transport agents. Aside from that, there have been some defects found in intrinsic TMDC flakes obtained from CVT-grown crystals. For instance, when TMDC flakes were used as gate material in the field-effect transistor (FET), these exfoliated flakes of TMDC showed either n-type or p-type behavior [129]. The film’s surface is homogeneous in the CVT process because it requires long synthesis time and high temperature. The distribution of both atoms Mo and W in CVT-grown ${Mo}_{x}W_{1-x}S_{2}$ alloy is uniform, but there is an in-plane gradient of Mo and W on the CVD-grown ${Mo}_{x}W_{1-x}S_{2}$ alloy [82]. The uniform distribution and density of substitutional point defects are finely tuned in the CVT process by loading the reaction chamber’s appropriate initial reactants. Moreover, bandgap engineering and doping in the materials can be performed in the CVT process [2]. Nevertheless, additional research needs to be conducted by introducing new metals and chalcogen atoms as dopants in the bulk and monolayer TMDC materials. The dominant defects observed in metal–organic chemical-vapor-deposited (MOCVD) TMDC films are chalcogen vacancy and substitutional defects. For example, the Se vacancies and substitution of W atom by Se atom are found in the epitaxially grown WS_2_ film on graphene. However, the defect density can be reduced (below 10^12^ cm^−2^) by annealing of as-prepared WS_2_ film in the presence of H_2_Se at 800 °C, as demonstrated in Figure 13b [53]. Another important technique to reduce defects during the synthesis process is the flux control method. In this method, the defect density is reduced substantially, and it is below 10^12^ cm^−2^. The comparison of defect density between CVT-grown and flux control methods is depicted in Figure 13c [53]. However, the charge impurities on the substrate surface introduce extrinsic disorder in the monolayer TMDC film. Two engineering approaches can reduce this extrinsic disorder from the film surface. One approach is the passivation of the substrate surface by a high-k dielectric material. This high-k passivation layer suppresses the charge impurities because the high-k dielectric constant reduces the effective Coulomb interaction between the TMDC film and substrate surfaces; thus, the substrate charge scattering along the cross-section reduces proportionally [130]. Another approach is the encapsulation of either side or both sides of an as-prepared TMDC film by hexagonal boron nitride (h-BN), as shown in Figure 13d. This heterostructure reduces charge disorder on the TMDC film by increasing conducting channel distance from the SiO_2_ substrate. The PL study confirms the reduced charge disorder density of TMDC films, as shown in Figure 13e [53].

### 5.2. Defect Engineering after Synthesis

Electrons in the monolayer MoS_2_ are very sensitive to impurities and to the interface. Therefore, structural, and interfacial defect engineering are equally essential to improve the performance of TMDC materials. Recently, some progress of ex situ defect engineering in 2D materials has been recognized. To begin with, electron beam irradiation (EBI) was performed for ex situ defect engineering of h-BN and graphene [2]. Although the application of EBI for defect engineering in 2D materials is costly and less effective for device application, nevertheless, this method has been applied in multi-layered or single-layered TMDC materials for generating chalcogen vacancy [131]. When mechanically exfoliated MoS_2_ sheet is placed under electron beam irradiation, then S vacancies are created. These vacancies are confirmed from aberration-corrected (AC) HRTEM images that contrast the Mo and S sub-lattice differences shown in Figure 14a [132]. This defect engineering might reduce contact resistance at the interface of metal–TMDC. The pristine and transition metal dichalcogenide interface introduces wide tunnel barriers due to the van der Waals (vdW) gap between metal and TMDC materials, resulting in higher contact resistance. The interfacial gap occurs due to the unavailability of overlapping free orbitals in TMDC materials. However, defect engineering, especially chalcogen vacancies, reduces metal–TMDC interfacial distance. The surrounding atoms of any chalcogen vacancy have unsaturated orbitals that create bonds with reacting elements. These unbounded orbitals overlap with metal atoms approaching the interface, enhancing the metal–TMDC bond. As a result, the metal–TMDC gap distance is reduced at the interface. Also, electron density at the overlapping orbital regions of metal–TMDC increases, reducing contact resistance barriers at the interface. This defect engineering technique is more suitable for Au, Cr, Pd, and Ni metal contact with MoS_2_, MoSe_2_, WS_2_, and WSe_2_ [18]. Interestingly, the S vacancies are mobile upon plasma irradiation, and they migrate and agglomerate along vacancy lines represented in Figure 14b [2]. In the defect engineering of TMDC materials using plasma, the energetic ions from plasma irradiation react with the material surface and change their properties. Either the ions strike the atoms and make a structural vacancy on the surface or form substitutional or adatoms impurities at the vacancy sites.

Surface engineering using plasma irradiation can be tuned effectively because the plasma source parameters are controllable and free from contaminations due to the dry environment in the reaction chamber [131]. Mechanically exfoliated MoS_2_ flakes, exhibiting n-type behavior, have S vacancy. Additionally, the Ar plasma creates S vacancies in the TMDC flakes. Contrastingly, these S vacancies of monolayer TMDC are repaired by oxygen plasma that introduces foreign oxygen atoms on the S vacancy sites and creates O-Mo bonds [133]. Also, substitutional O adatoms induce a charge on the S vacancy site, leading to p-doping, which is confirmed from the PL study depicted in Figure 14c [131]. Additionally, PH_3_ plasma treatment of MoS_2_ induces p-type behavior [86]. Moreover, H_2_/He plasma introduces Se vacancy in the WSe_2_, and this Se vacancy induces n-type doping in this TMDC material [134]. Also, ripple formation and separation of TMDC layers are performed using CHF_3_ and SF_6_ plasma treatment [135]. Additionally, the phase change in TMDC materials can be performed by plasma treatment. For example, if the Ar plasma treats the 2H phase of MoS_2_, it converts the phase structure of MoS_2_ from 2H to 1T by sliding the top S-layer [131]. However, chemical treatment is comparatively better than plasma irradiation because chemically functionalized materials are more defect-free than plasma-treated surfaces. Sulfur vacancy is the most prominent intrinsic defect in MoS_2_, and this defect can be healed using thiol chemistry. The healing mechanism depends on two-step chemical reactions exhibited in Figure 15a [136]. The S defect is repaired by “(3-mercaptopropyl) trimethoxysilane (MPS)” by overcoming two energy barriers of 0.51 eV and 0.22 eV using heat treatment. Also, the MPS treatment of MoS_2_ reduces the charge impurities, small scattering, and improves the carrier mobility at low temperature. However, the mechanically exfoliated MoS_2_ flakes from bulk crystals have a higher sulfur vacancy. These defects are chemically active, especially hydrodesulfurization reaction.

The thiol group of the MPS molecules repairs the sulfur vacancy (SV). The MPS-passivated MoS_2_ surface is annealed at 350 °C, forming gas of MPS. The following reaction occurs:HS (CH_2_)3Si (OCH_3_)_3_ + SV → CH_3_ (CH_2_)_2_Si (OCH_3_)_3_

The thiol group in gaseous MPS is absorbed onto the SV through the sulfur atom. After that, its S-H bonds are broken, and an intermediate thiolate group is formed by releasing H that forms a bond with the nearest S atom of TMDC material. Later, the S-C bond breaks down, and the intermediate thiolate converts to trimethoxy (propyl) silane after hydrogenation [138]. The transmission electron microscopy images shown in Figure 15b indicate that the MPS treatment reduces the defect density on the MoS_2_ surface approximately four times compared to as-exfoliated MoS_2_ [136]. Another chemical reagent for sulfur vacancy healing is “poly (4-styrenesulfonate) (PSS)”. In this “sulfur vacancy self-healing mechanism (SVSH)”, PSS plays the catalyst role in this reaction. The sulfur adatoms cluster on the as-grown MoS_2_ surface, which fills the S vacancy by the guiding of hydrogenated PSS. “High angular annular dark field (HAADF) scanning transmission electron microscopy (STEM)” images presented in Figure 15c reveal that there are no mono-sulfur vacancies and S adatoms clusters on the PSS-functionalized MoS_2_ surface [84]. Moreover, the defects of TMDC materials can be repaired using ultraviolet ozone treatment exhibited in Figure 15d [137]. In this method, the released ozone (O) from oxygen molecules after exposing ultraviolet rays is a strong oxidant. This releasing oxidant oxidizes the TMDC materials such as WSe_2_ and makes a transitional metal–oxygen bond that suppresses chalcogen atoms’ vacancy. The resulting TMDC materials are the perfect hexagonal structure, but the p-type carriers are increased in the lattice structure [83]. Moreover, the laser rays in the presence of oxygen heal the Se vacancies in the WSe_2_ [85]. Also, ion irradiation is used to make defects in the single-layer MoS_2_. Shyam et al. reported the generation of point defects in the single layer of MoS_2_ using ion irradiation. Sulfur point defect formation was created using helium ion irradiation. Also, sulfur defects quantification was performed using angular detector of scanning transmission electron microscopy [139]. Additionally, thermal treatment is used for defect engineering of TMDC materials. In this healing process, the as-grown TMDC monolayer is annealed in the presence of oxygen molecules. These oxygen molecules absorb at the chalcogen site and cause transition metal–oxygen bonding, leading to the suppression of chalcogen vacancy [131]. Further development of defect engineering is needed to identify ways to control the TMDC layers, chemical functionalization, and improve the van der Waals system so that TMDC materials can contribute extensively to the electronics industry.

## 6. Impact of Defects on the TMDC Materials’ Properties

This section focuses on the impacts of defects on TMDC materials, including electronic, optical, and magnetic properties. Sometimes, they enhance the properties, and sometimes they are detrimental. A detailed discussion is undertaken below.

### 6.1. Impact on Electronic Properties

In the TMDC materials, zero-dimensional defects such as chalcogen vacancies introduce free electrons in the lattice structure, changing the materials’ conductivity to n-type. For instance, S deficiency in MoS_2_ reveals the n-type transfer characteristic shown in Figure 16a [140]. On the other hand, S-rich or Mo-deficient, MoS_2_ exhibits p-type behavior represented in Figure 16b [140]. Regardless, both types of carrier behavior are observed in the same MoS_2_ sample but in different regions [140]. Therefore, additional improvements are required to maintain the desired carrier type of TMDC materials’ flakes during the synthesis process. The mono-S vacancy has two impacts on the density of the state. First, defect states are formed close to the conduction band minimum (CBM) edges. These defect states are formed due to the unsaturated charge in 4d orbitals of Mo atoms that make dangling bonds. Another effect is the shallow state change close to the valence band maximum (VBM). The state optimization occurs near the valence band due to the S vacancy, leading to a decrease in orbitals hybridization of S (3p) and Mo (4d) atoms. However, in the bandgap, transition metal vacancies produce defect states that pin the Fermi level, detrimentally impacting the device performance. Indeed, Mo vacancies form defect states near to the VBM. These defect states extend to the mid-gap due to the dangling bonds of the surrounding six S atoms. These defects act as trapping centers of other impurities, leading to the additional Fermi level pinning [141]. Additionally, S vacancies induce localized charges around it. At low carrier density, these localized electrons at elevated temperature transport to the neighboring defect states, or at low temperature they transport to the variable-range defect states [24]. However, when the S adatoms are adsorbed on the S sites, they form localized gap states adjacent to the VBM; thus, bandgap narrowing occurs. The reason for the band narrowing is the hybridization of s-orbital of S atoms with the d-orbital of Mo atoms [141]. Moreover, foreign atoms can modulate the bandgap of TMDC materials. For example, the doping variations in the monolayer ${Mo}_{x}W_{1-x}S_{2}$ and ${Mo}_{x}W_{1-x}{Se}_{2}$ alloys significantly modulate the bandgap, confirmed from the PL study shown in Figure 16c [56]. Due to the bowing effect, this bandgap does not change linearly according to the stoichiometry [56]. Nb-doped MoS_2_ shows a p-type transitional characteristic that has been theoretically proven [65].

Also, DFT calculation reveals that rhenium (Re) is an n-type dopant [65]. It is also experimentally verified that Re is an n-type dopant. In electrocatalytic reaction, the onset potential is reduced for low Re doping. Also, low Re doping improves overall hydrogen evolution reaction (HER) performance [31]. The direct bandgaps of ReS_2_ and ReSe_2_ decrease under applied pressure. The band gap properties of these transition metal dichalcogenide materials are different than other transition metal dichalcogenides, which show increasing bandgaps under pressure. The sensitivity of the bandgap to pressure could enable new types of pressure-sensitive optical devices [36]. Furthermore, 1D defects such as grain boundaries, line defects, and inversion domains greatly influence the electronic properties of TMDC flakes. When the S line vacancy encompasses the inversion domain, it changes the inversion domain edges’ stoichiometry, leading to defect states formation in the bandgap. The transition metals are accountable for the formation of mid-gap states [48]. However, the simultaneous presence of semiconducting (2H) and metallic (1T) phases in MoS_2_ monolayers has a significant impact on the two-dimensional lateral heterojunctions. Introducing sulfur vacancies (V_S_ and V_2S_) at the interface introduces new defect states in close proximity to the Fermi level, thereby enhancing electron density locally. The impact is particularly notable for divacancies aligned parallel to the interface. Molybdenum vacancies (V_Mo_) and vacancy complexes (V_Mo+3S_, V_Mo+6S_) give rise to resonant defect states near the band edges originating from Mo 4d orbitals, thereby boosting carrier injection and transmission at the interface. Anti-site defects, with a focus on substitutional defects like 2S at the Mo site (2S-Mo), generate localized mid-gap states through Mo-S hybridization, while other anti-sites have minimal effects. Among these defects, Mo vacancies induce the most significant modification in the interfacial electronic structure and density of states near the Fermi level [47]. Likewise, epitaxially grown monolayer MoSe_2_ has Se deficiency that leads to a twin grain boundary. Grain boundaries, especially non-zigzag-oriented mirror twin boundaries (MTBs), introduce electronic defect states in the bandgap that reduce the bandgap locally by up to 1 eV. This degrades the electronic quality and homogeneity of CVD-grown MoSe_2_ sheets [25]. The first-principles calculations predict that the conductivity is increased due to the increase in localized charges in the bandgap [32]. However, grain boundary can be p-type or n-type depending on the terminated atomic structure of the grain. 

As a result, locally doped materials are formed. Bandgap variation is observed between the two neighboring grain boundaries shown in Figure 17a [2], which is confirmed from the scanning tunneling spectroscopy (STS) depicted in Figure 17b [2]. This bandgap variation suggests that grain boundaries locally modify the electronic structure of the TMDC materials. CVD-grown grain boundaries modulate the in-plane carrier conductivity of monolayer MoS_2_, but MOCVD-grown grain boundaries exhibit little carriers’ hindrance in the MoS_2_ films. Nevertheless, sufficient improvements in the synthesis method are to be performed to produce large-scale single-crystalline TMDC films [2]. Recently, polycrystalline TMDC films have been used in memristor applications because the memristors’ resistance emerges from the contacts of the neighboring grains modulated by the external voltage represented in Figure 17c [37]. Also, line defects reduce the rotational symmetry but increase the in-plane anisotropy in the quantum conductance [41]. Hetero-interfaces have gained attention in the research community, and lateral heterostructures can be synthesized by epitaxial growth. The hetero-interface between the two different TMDC materials is another one-dimensional defect that forms a p–n junction. For instance, epitaxially grown MoS_2_/WS_2_ and MoSe_2_/WSe_2_ in-plane lateral heterostructures form 1D interfaces that are a p–n junction [42,61]. The monolayers of lateral stacking of the TMDC materials interact with each other by van der Waals forces. These van der Waals interfaces act as a 2D defect, impacting the electronic properties of TMDC materials. When the layers of single-phase TMDC materials are reduced to one, then the bandgap of these materials transforms from direct to indirect transition. For example, the single layer of WSe_2_ or MoS_2_ has a direct bandgap, but, beyond the monolayer, these TMDC materials show indirect bandgap transition [2]. However, the DFT calculations predict that the hetero-bilayers’ lateral structure of TMDC materials can be direct bandgap semiconductors, but the bandgap energy of this heterostructure is lower than each constituent part [61]. On the other hand, the experimental study demonstrates that the bilayer heterostructures of MoS_2_-WS_2_ and MoSe_2_-WSe_2_ exhibit indirect bandgap, and even mono-bilayers of these heterostructures show an indirect transition. Nevertheless, bilayer heterostructures display higher rectification current, good photovoltaic response, and improved short circuit current (~10^3^) [17]. These electronic properties of heterostructures depend on individual monolayers and interlayer spacing. Therefore, the bilayers’ heterostructures are the right candidate for optoelectronic device applications. Additionally, after applying these bilayer heterostructures, such as MoS_2_-WSe_2_, in the tunneling diode and photodiodes, new functionalities have been obtained. So, more efforts have to be devoted to investigating further to clarify the contradictory results between theoretical and experimental works of bilayer heterostructures.

### 6.2. Impact on Optical Properties

The semiconducting TMDC materials’ optical properties have a direct relationship with the band structure and the excitonic transition. From the electronic properties of TMDC materials, the dopant induces band gap variation. Likewise, the relation between bandgap and disorder is confirmed from the photoluminescence spectroscopy. When the exfoliated MoS_2_ flakes are irradiated by control plasma, then bi-sulfur vacancies are produced. The observed PL peak energy from the sample is shifted below the bandgap value of monolayer MoS_2_ shown in Figure 18a [44]. The shifting of PL peak to the low-energy end happens due to the neutral excitons binding with the plasma-created defects. In other words, the produced defect states in the band gap reduce the treated sample’s overall band energy. Additionally, when the exfoliated monolayer MoS_2_ flake is annealed at ~100 °C, below the decomposition temperature (~600 °C), S vacancies are created, and the PL peak shifts to 1.78 eV, as represented in Figure 18b [44]. Intrinsic defects like sulfur vacancies act as electron donor states, resulting in shallow trap states that lead to slower photoresponse. Also, the interface like MoS_2_/SiO_2_ introduces trap state that impacts photogating effect of the device [49]. On the other hand, extrinsic defects like adsorbed molecules O_2_ and H_2_O remove electrons from MoS_2_ and act as trap states, slowing photoresponse. The electron withdrawal and trap states caused by adsorbates suppress the electronic conductivity within the MoS_2_ layer. This reduces the efficiency of photogenerated carrier collection, and adsorbed molecules under light dominate photocurrent dynamics. The defect density in MoS_2_ dominates adsorbate molecules on the surface [49]. The details are shown in Figure 18f. However, van der Waals hetero-epitaxy of vertically aligned TMDC layers is possible due to each layer’s free dangling bonds. This atomically thin interface between two adjoining layers is 2D in nature, and this vdWs heterostructure reveals the novel properties that are not present in each TMDC flake. This heterostructure tunes the bandgap, consequently altering the excitonic transition in the hetero-bilayers. The bandgap of monolayer MoS_2_ and WSe_2_ is ~1.87 eV and ~1.64 eV, respectively, but, when they are mechanically stacked one after another, the PL peak position is shifted to ~1.50–1.56 eV, as depicted in Figure 18c [2]. This bandgap lowering occurred due to the excitonic binding to the 2D interface of this vdW hetero-epitaxy. Additionally, the hetero-bilayers’ band alignment is type-II, and the transition of excitons is indirect [2]. Also, the point defects identified as molybdenum (Mo) vacancies in MoSe_2_ introduce electronic defect states in the bandgap, reducing the bandgap by ~0.3 eV [25]. Usually, the crystal structure and lattice constant of TMDC materials could be similar to each other. Therefore, the in-plane vertically aligned heterostructures of MoS_2_/WS_2_ and MoSe_2_/WSe_2_ are formed using vapor phase deposition with less lattice mismatch.

A large portion of the excitonic recombination happens at the heterojunction interface of TMDC materials, resulting in higher-intensity light emission than the individual TMDC flakes shown in Figure 18d [2]. However, the energy of radiated light from the hetero-interface lies in between the band gaps of two interfacial layers [2]. Also, because of the absence of inversion symmetry (IS) in the single-layer TMDC films, non-linear optical properties are generated, such as second harmonic generation (SHG). Several factors influence SHG signals’ intensity, such as grain boundaries, crystal orientations, and excitonic wavelength. For example, in the chemical-vapor-deposited polycrystalline MoS_2_ flakes, the SHG signal’s intensity is reduced at the grain boundary due to destructive interference from the adjacent grains. On the other hand, the SHG signal intensity is increased at the Mo-zz-terminated edges. However, the ideal stacking 2H Bernal configuration of TMDC’s even number of layers does not exhibit the SHG signal. On the contrary, the even number of TMDC stacking layers with non-ideal atomic configuration shows the SHG signal because of the lack of inversion symmetry. The tilting angle between the adjacent layers modulates the interference of SHG signals produced from each stacking layer shown in Figure 18e [2]. The optical properties with in-plane anisotropy in ReS_2_ and ReSe_2_ remained unchanged up to pressures of 20 kbar, indicating the possibility of designing optoelectronic devices that can maintain performance under mechanical strain or pressure fluctuations. The optical transitions in ReS_2_ and ReSe_2_ remain relatively stable across a broad temperature range [36]. This implies the potential for devices to withstand variations in temperature. However, further research is needed to investigate other non-linear optical properties that exist in this heterostructure. Additionally, the foreign-atoms-doped TMDC heterostructures should be investigated to reveal a new dimension in the optoelectronic area.

### 6.3. Impact on Magnetic Properties

Natural TMDC materials exhibit nonmagnetic behavior, i.e., no magnetic moments within them. However, various structural defects, including vacancies, adatoms adsorption, and edge deformation, cause a magnetic moment in the TMDC materials. Vacancies introduce magnetic moments in the TMDC materials by transferring charges between transition metals and chalcogen atoms located near the vacancies. Additionally, the dangling bonds in the distorted MoS_2_ lattice produce spin polarization due to the enormous charge variation around the defects. Moreover, the adatoms generate a magnetic moment in the TMDC lattice structure because of the partially filled d-orbital and empty s-orbital [67]. Additionally, the atoms at the TMDC-terminated edges play a vital role in the magnetization because the truncated atoms do not maintain the same bulk stoichiometry; thus, disoriented spin distribution occurs. However, the edge reconstructions can be Mo–Mo or S–S bonding, causing a ${Mo}_{x}S_{y}$ cluster. In this cluster, Mo atom is responsible for the magnetic moment because of the d-orbital is partially filled. Additionally, the trigonal prismatic and octahedral coordination exhibit magnetic properties due to the spin polarization of d^2^ electrons. Recently, first-principles calculations predicted that ${the\;Mo}_{n}S_{2n}$ cluster shows magnetic polarization. For example, the energy difference between magnetic molecular cluster ${Mo}_{6}S_{12}$ and nonmagnetic isomer is higher (0.94 eV) [23]. It is also predicted from the calculation of spin polarization that a one-dimensional edge defect introduces non-zero magnetic moment in the MoS_2_ or WSe_2_ due to the unpaired electrons shown in Figure 19a [2]. However, the first-principles calculations predict that weak ferromagnetism is observed in the MoS_2_ nanosheets, nanoribbons, and clusters [26]. Various types of magnetism, e.g., ferromagnetism, paramagnetism, and diamagnetism, are found experimentally in the mono- and few-layered TMDC materials. Nevertheless, the TMDC materials cannot be used in spintronics because of these materials’ low saturation magnetization. To enhance TMDC materials’ magnetism, such as MoS_2_, additional efforts have been undertaken to change these materials’ microstructure. Recently, some methods have been demonstrated to change the structural properties of these materials. Ion irradiation is used to create structural defects such as atomic vacancies and displacement that increase ferromagnetism. Also, the formation of anti-site defects in the MoTe_2_ or MoSe_2_ during growth makes these suitable magnetic materials. The magnetism observed due to the interaction between local magnetic domains appears due to electron spin polarizations. Magnetism emerges when the anti-sites’ defect densities are around 0.4% and are randomly distributed over the crystal. Low densities of these defects in TMDCs lead to self-organized magnetism, and controlling defect densities could tune magnetic properties [33]. Additionally, decreasing the thickness of the MoS_2_ increases the magnetization represented in Figure 19b [23]. Because the bulk and thick films of the TMDC materials have low prismatic edges due to secondary nucleation, the increased size of the secondary crystals leads to a decrease in magnetization [23]. Moreover, the first-principles calculation predicts that, when the dislocation cores create angular alignment in a specific grain boundary (GB), this GB possesses magnetism due to the partial occupancy of localized electronic state. Also, the DFT calculation of MoS_2_ relates the magnetism of grain boundaries with the tilt angle. According to this theoretical prediction, the grain boundary’s tilt angle introduces two magnetic orders in the TMDC nanosheets. If the tilting angle of grain boundaries and dislocation orientation of TMDC materials are <47° and 5|7 ring, respectively, they show ferromagnetic characteristics. On the other hand, the grain edges with tilt angle >47° and 4|8 dislocation cores show the antiferromagnetic behavior depicted in Figure 19c.

The main reason for changing the magnetic behavior with the tilt angle is the variation in the magnetic moment per unit length, which is increased with an enlarging tilt angle [45]. The DFT calculations indicate that, when Ti, V, Cr, Ni, Mn, and Zn transitional metals (3d) are doped as substitutional atoms, they introduce the TMDC materials’ magnetic order [2]. However, a theoretical study reveals that Mn doping introduces long-range ferromagnetic behavior in the host TMDC materials and makes these materials retain the magnetic properties until Curie temperature. As a result, the Mn-doped MoS_2_ can be used as a “diluted magnetic semiconductor” in several applications. Recently, Mn has been doped in MoS_2_ using the CVD process, where Mn_2_(CO)_10_ is used as a Mn precursor. Additionally, further improvements are needed to enrich the magnetism of TMDC materials [2]. (See Table 3).

## 7. Summary and Outlook

TMDCs are promising 2D materials with rich novel physical characteristics that expand their domain in different nanoscale applications, such as electronics, optoelectronics, and spintronics. However, the current electronics industry is vastly dominated by group IV semiconductors such as silicon and germanium. Nevertheless, silicon-based devices will reach their end soon because of the physical limitations of the current technology. Thus, emerging two-dimensional TMDC materials are promising either as gate oxide or as a substrate to fabricate nanoscale devices. Additionally, the future electronics devices will evolve as foldable because of their conformability, higher durability, space efficiency, and light weight. The conventional group IV semiconductor substrates are rigid; thus, they are not suitable for flexible electronics. On the other hand, TMDC materials’ substrates are stretchable, and the electronic and optical properties change little with bending. Therefore, these materials are suitable candidates for stretchable electronics. Moreover, the excellent chemical stability and high ratio of surface to volume make TMDCs promising for power storage devices and catalytic applications because of their optimized surface functionalization. Aside from that, there is good adhesion compatibility of TMDCs with organic materials that will lead to the application of these 2D materials in organic electronics, such as organic light-emitting diodes (OLEDs), organic photovoltaic cells (OPVs), and organic field-effect transistors (OFETs). Also, TMDC materials can be used in logic circuits because of their high on/off ratio. Additionally, atomically thin TMDC materials have promising applications in biophysics. Rapid high-resolution DNA sequencing can be performed using TMDC materials because of their atomically thin nanopore membrane. Nevertheless, these promising applications of TMDC materials face challenges because of their structural defects. Most of the intrinsic defects, such as S or Mo vacancies, arise from various extracting ores or synthesis processes, including CVD, ALD, MOCVD, and PVD. Various synthesis processes introduce different defects on the TMDC film surface. In contrast, extrinsic defects are either unwanted or intentionally created. Most of the unwanted extrinsic defects are introduced from the environment, such as the film surface’s oxidation leading to dangling bonds. On the other hand, intentionally produced defects are formed by various engineering methods. Defect engineering plays a vital role in tuning TMDC materials’ properties by structural modification, either via ex situ or in situ processes. In the in-situ defect engineering process, the defect density can be optimized by controlling various parameters of synthesis tools, the ambient temperature and pressure, and the surface passivation of substrates before deposition. On the other hand, various defect engineering methods are used in the ex-situ process, including ion beam irradiation, plasma treatment, surface functionalization of TMDC materials using thiol chemistry, and ozone treatment. These ex-situ processes either increase or decrease the defect density of TMDC materials. Sometimes, defects are not detrimental; instead, they play a beneficial role in device applications. For example, the substitutional dopants in the TMDC materials control the metal–TMDC semiconductor interface’s work function. Also, defects in the TMDC materials act as optically active sites. When these optically active sites are controlled and coupled precisely with charges in the interface of a van der Waals heterostructure, then TMDC works as a quantum emitter that controls information processing. Further investigations of TMDC materials needed to determine their usability in vast applications are outlined here. Firstly, we investigate how to improve the defect-free crystals during synthesis. For example, more efforts are necessary in TMDC synthesis using MOCVD and CVD methods for increasing the grain size so as to reduce the grain boundary, which will help regarding the wafer-scale production of TMDC films needed for commercialization in electronics applications. Additionally, the substrate’s surface passivation has a massive influence on defect-free TMDC film growth because the charged impurities on the defect-induced surface alter the periodic order of the deposited films. Furthermore, h-BN and high-k dielectric materials are used as the surface passivation layer. However, the surface of high-k materials has dangling bonds, which impede the ideal heterostructure formation of TMDC and dielectric materials. A few studies have been conducted to identify substrate passivation’s impact using high-k materials on film growth. More investigations on the substrate’s surface passivation using other innovative materials are needed. Secondly, more emphasis is needed for defect creation and utilization of TMDC materials because the manipulation of defects at the atomic scale is not clearly documented. Also, the basal plane’s chemical functionalization is needed to know and to study the impacts of the defects on the mechanical and thermal properties of TMDC materials. On the other hand, defects are beneficial because they tune various properties of these materials. For instance, the interface of the heterostructure of several TMDC materials’ layers forms 2D defects. Such 2D defects at the interface exhibit novel characteristics, paving the way to use these heterostructures in innovative applications. However, the hindrance to this aspect is that it is unknown how to control the defect formations at a specific place of the TMDC films as well as the migration of defects within these films. Also, using defect engineering in the TMDC materials, it is still impossible to fabricate a single-photon photodetector with high electroluminescence. Additionally, more investigations are needed to identify the impacts of defect creation on the materials’ electronic and optical properties. Thirdly, the post-synthesis defect engineering methods have to improve to heal the defects from the surface because, during synthesis at high temperature, defect formation is a common scenario. There are some common post-synthesis defect engineering methods available, including heat treatment and thiol chemistry. Nevertheless, they do not entirely remove defects from the surface. Therefore, it is necessary to identify more suitable approaches to heal defects after synthesizing TMDC materials. Finally, it is more important to improve the intrinsic crystal quality of 2D materials to better understand structural defects to achieve large-scale applications in various fields.

## 8. Conclusions

Transition metal dichalcogenides are novel and potential materials because of their layered dependent properties. These unique physical properties facilitate applications in various cutting-edge fields. However, structural defects impede the successive progression of TMDC materials. The dominant defects in these TMDC materials are point defects, line defects, and grain boundaries. Most of these defects occur during the synthesis process. It is difficult to achieve wafer-scale production of TMDC films using the existing methods because these deposition methods produce more grain boundaries on the film surface. However, the conventional defect engineering methods reduce the amount of defect density on the film surface less. So, the current methods are not good enough for defect engineering in various applications. Therefore, further research is needed to develop the existing synthesis tools and defect engineering methods to solve the structural defect problems on the TMDC surface and make these materials useable for future applications.

## Figures and Tables

**Figure 1 nanomaterials-14-00410-f001:**
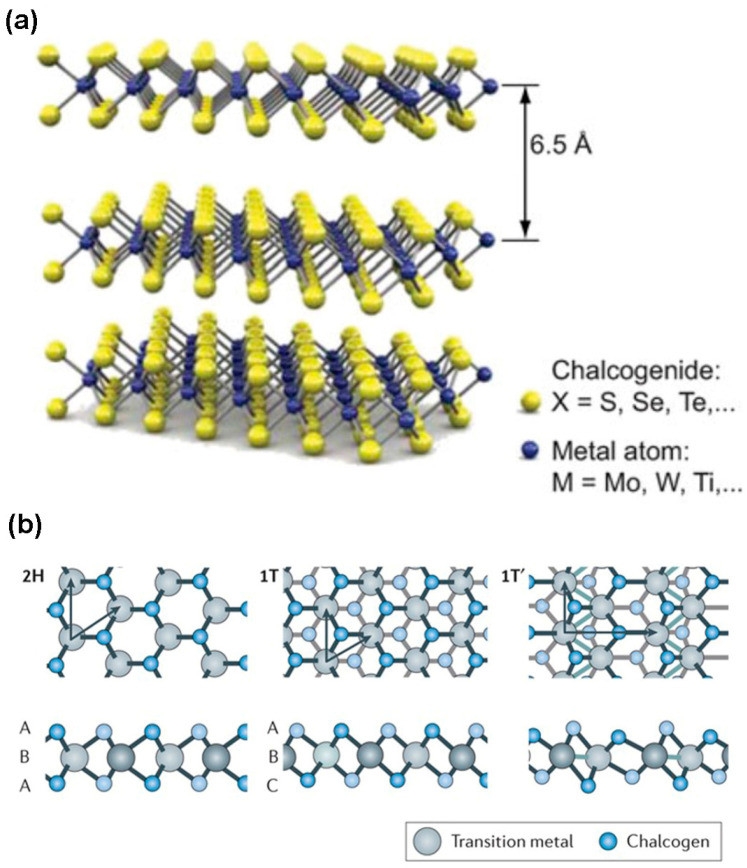
(**a**) The general structure of TMDC materials Reprinted with permission from Ref. [19]. 2015, New York, Cambridge Univ. Press (**b**) common phases of TMDC materials. Reprinted with permission from Ref. [1]. 2015, Macmillan Publishers Limited, part of Springer Nature.

**Figure 2 nanomaterials-14-00410-f002:**
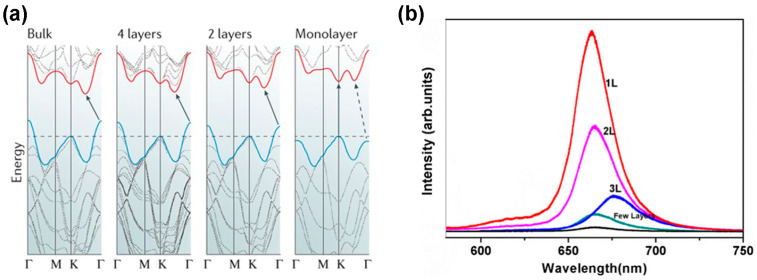
(**a**) Graphical representation of thickness-dependent band gap transition of TMDC materials from first-principles calculation. Reprinted with permission from Ref. [1]. 2017, Macmillan Publishers Limited, part of Springer Nature (**b**) layer-dependent photoluminescence spectra of MoS_2_. Reprinted with permission from Ref. [20]. 2018, Taylor & Francis Group, LLC.

**Figure 3 nanomaterials-14-00410-f003:**
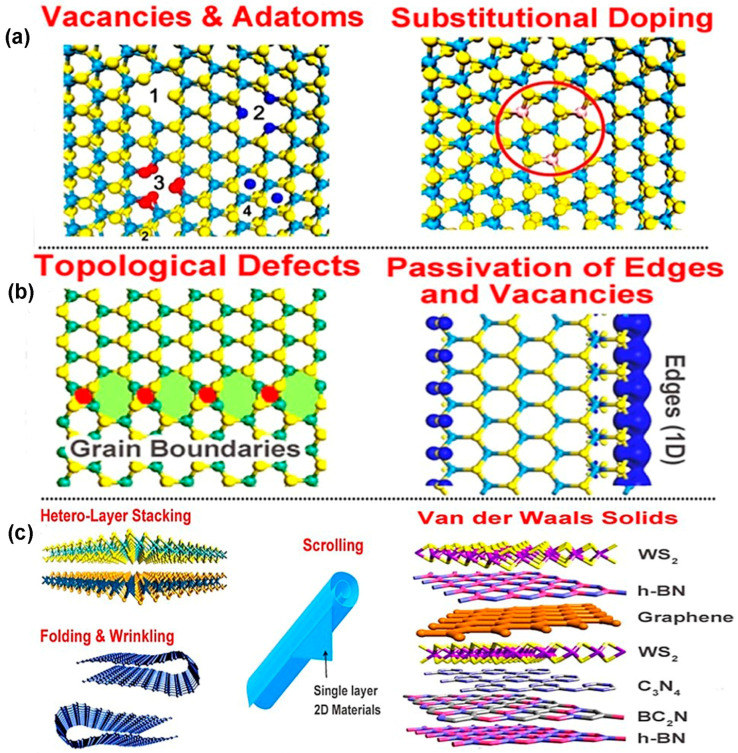
Classification of 2D transitional metal dichalcogenide materials’ defects: (**a**) zero-dimensional (0D) defects; (**b**) one-dimensional (1D) defects; and (**c**) two-dimensional (2D) defects. Reprinted with permission from Ref. [2]. 2016, IOP science.

**Figure 4 nanomaterials-14-00410-f004:**
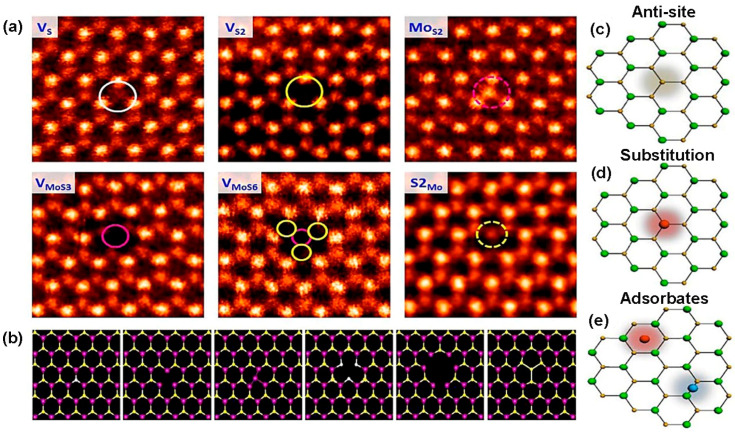
Zero-dimensional defects: (**a**) structural point defects of MoS_2_; (**b**) DFT calculation of point defects of MoS_2_. Reprinted with permission from Ref. [52]. 2013, American Chemical Society (**c**) anti-site formation; (**d**) substitution of dopant atoms; and (**e**) foreign atoms adsorption on the TMDC surface. Reprinted with permission from Ref. [53]. 2019, Nature.

**Figure 5 nanomaterials-14-00410-f005:**
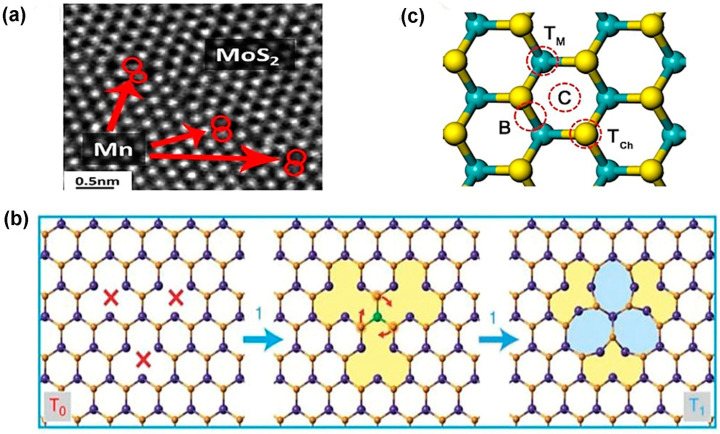
(**a**) ADF-STEM image of Mn-doped MoS_2_. Reprinted with permission from Ref. [68]. 2015, American Chemical Society (**b**) the formation mechanism of rotational defects. Reprinted with permission from Ref. [66]. 2015, Nature Research; and (**c**) the schematic diagram of adatoms adsorption on four sites of TMDC lattice. Reprinted with permission from Ref. [2]. 2016, IOPscience.

**Figure 6 nanomaterials-14-00410-f006:**
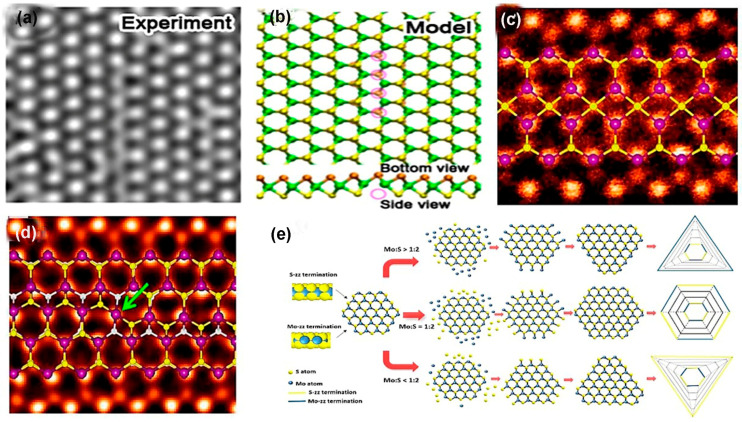
One-dimensional defects: (**a**) HRTEM images of single S and double S vacancies. Reprinted with permission from Ref. [71]. 2013, American Physical Society (**b**) theoretical model of line vacancy of TMDC materials. Reprinted with permission from Ref. [71]. 2013, American Physical Society; annular dark field images of 60° grain boundaries; (**c**) 4|4P; (**d**) 4|4E. Reprinted with permission from Ref. [52]. 2013, American Chemical Society; and (**e**) schematic illustration of shape evaluation depends on the ratio of Mo and S. Reprinted with permission from Ref. [72]. 2014, American Chemical Society.

**Figure 8 nanomaterials-14-00410-f008:**
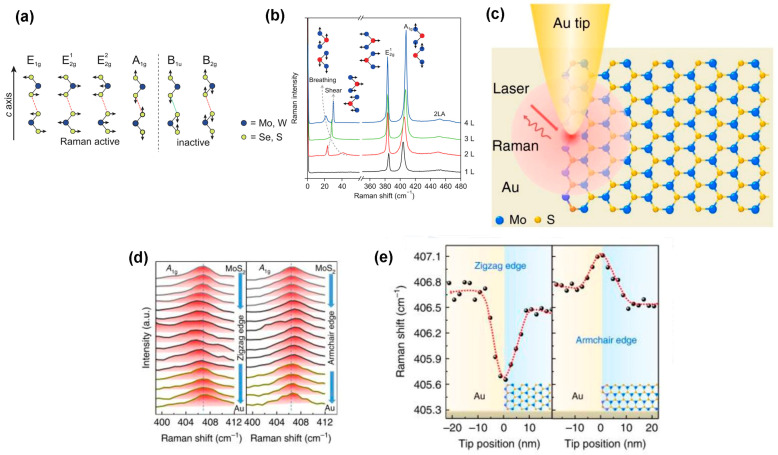
(**a**) Schematic representation of Raman modes of TMDC materials. Reprinted with permission from Ref. [89]. 2013, Optical Society of America; (**b**) representation of Raman spectra of MoS_2_ to determine the number of layers. Reprinted with permission from Ref. [90]. 2015, Korean Society of Microscopy; (**c**) the tip-enhanced Raman spectroscopy (TERS) configuration of Au-coated AFM tip; (**d**) line spectra of TERS for zigzag edge (left) and armchair edge (right) under A1g mode of AFM; and (**e**) variation in peak position according to the AFM tip positions. Reprinted with permission from Ref. [96]. 2019, Nature Research.

**Figure 9 nanomaterials-14-00410-f009:**
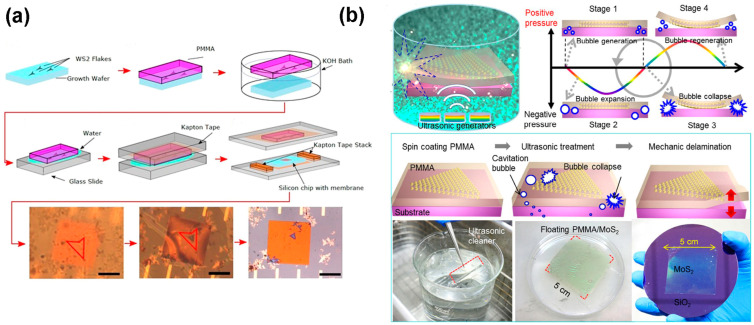
Schematic diagram of the TMDC flakes exfoliation from the growth substrate: (**a**) wet etching process. Reprinted with permission from Ref. [108]. 2017, Nature Research; (**b**) physical exfoliation process. Reprinted with permission from Ref. [109]. 2015, Springer.

**Figure 10 nanomaterials-14-00410-f010:**
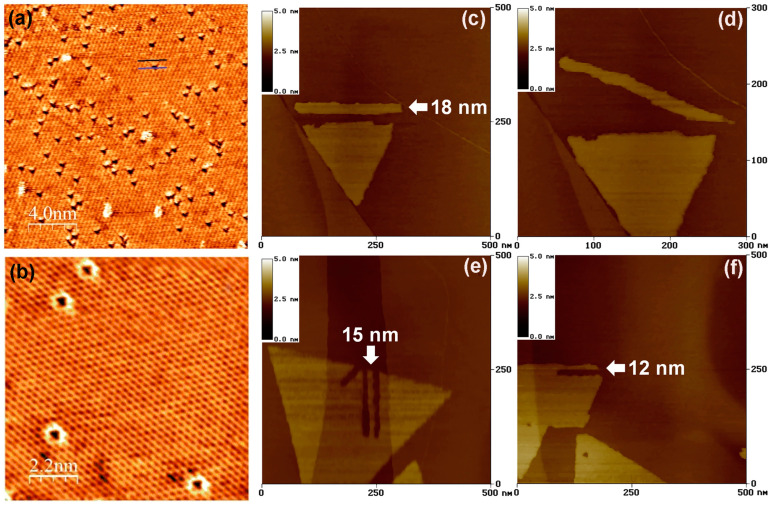
Atomic resolution of STM imaging of (**a**) triangular point defects; (**b**) circular point defects of monolayer MoS_2_. Reprinted with permission from Ref. [117]. 2016, Nature Research; and (**c**–**f**) MoS_2_ nanoribbon cut by STM nanolithography. Reprinted with permission from Ref. [119]. 2016, Elsevier.

**Figure 11 nanomaterials-14-00410-f011:**
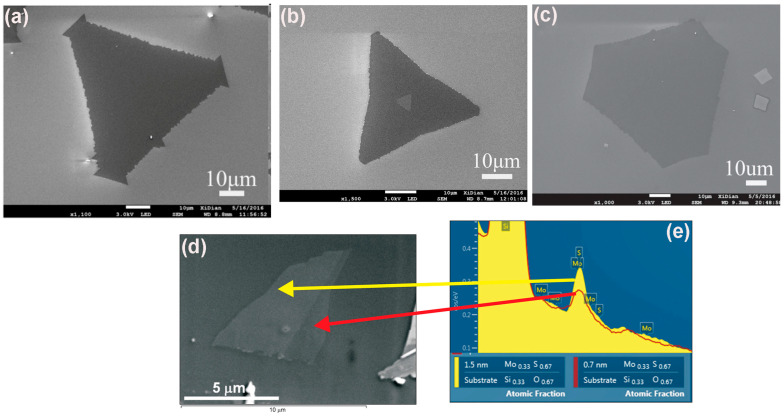
SEM micrograph of MoS_2_ flakes for different ratios of Mo and S: (**a**) triangular; (**b**) truncated triangular; (**c**) hexagonal. Reprinted with permission from Ref. [120]. 2017, IOPScience; (**d**) MoS_2_ flakes on the Si substrate; and (**e**) corresponding EDS spectra and layer thickness information. Reprinted with permission from Ref. [121]. 2015, Korean Society of Microscopy.

**Figure 12 nanomaterials-14-00410-f012:**
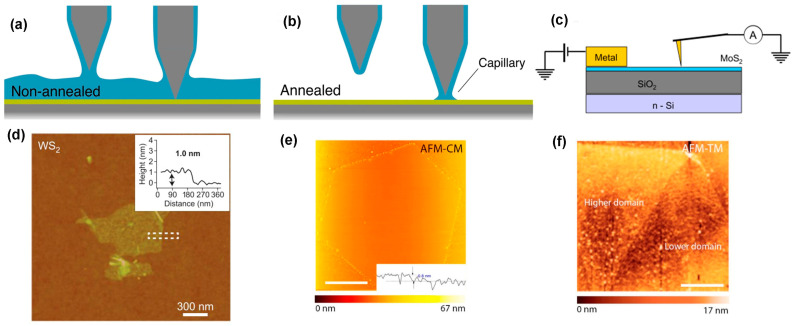
Schematic of AFM tip sample force variation: (**a**) before annealing; (**b**) after annealing. Reprinted with permission from Ref. [122]. 2017, Nature Research; (**c**) current–voltage (I-V) measurement on the MoS_2_ surface. Reprinted with permission from Ref. [124]. 2016, Elsevier. The height profile of (**d**) WS_2_ on the silicon substrate. Reprinted with permission from Ref. [123]. 2013, Nature Research. AFM topographic images of WS_2_ monolayers: (**e**) tapping mode with domain contrast, and (**f**) contact mode. Reprinted with permission from Ref. [125]. 2017, American Chemical Society.

**Figure 13 nanomaterials-14-00410-f013:**
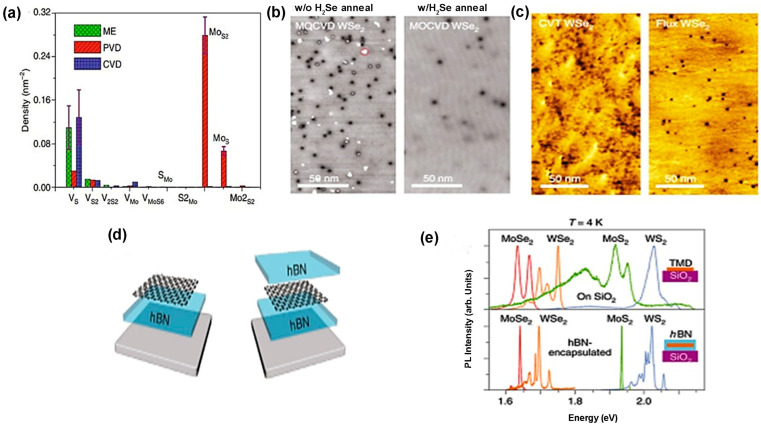
In situ defect engineering method: (**a**) comparison of defect densities in various fabrication methods; (**b**) reduction in defect density of WSe_2_ film using MOCVD method; (**c**) comparison of defect density between CVT and flux control methods; (**d**) encapsulation of substrate surface using h-BN to reduce defects’ density; and (**e**) comparison of photoluminescence study among TMDC materials with and without encapsulation by h-BN. Reprinted with permission from Ref. [53]. 2019, Nature.

**Figure 14 nanomaterials-14-00410-f014:**
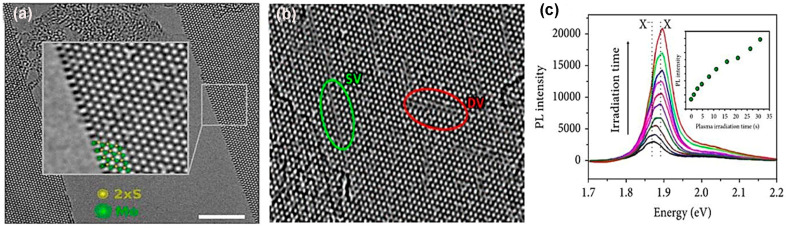
(**a**) Aberration-corrected (AC) HRTEM images for confirming S vacancy. Reprinted with permission from Ref. [132]. 2012, American Physical Society; (**b**) migration of S line vacancy after plasma treatment ] Reprinted with permission from Ref. [71]. 2013, American Physical Society; and (**c**) photoluminescence study of TMDC sample for confirming O atoms adsorption on its surface. Reprinted with permission from Ref. [131]. 2014, American Chemical Society.

**Figure 15 nanomaterials-14-00410-f015:**
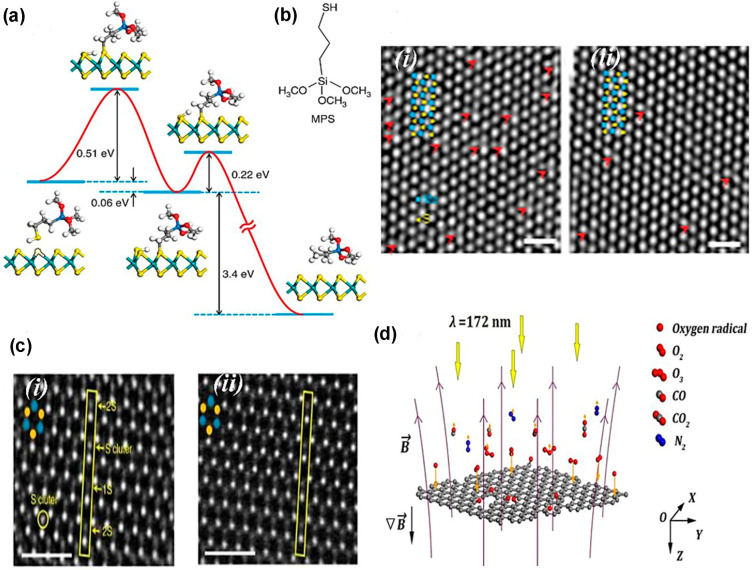
Ex situ defect engineering methods: (**a**) mechanism of S healing using thiol chemistry; (**b**) TEM images of MoS_2_ sample surface (i) before thiol treatment and (ii) after thiol treatment. Reprinted with permission from Ref. [136]. 2014, Nature Research; (**c**) ADF-STEM images of self-healing S vacancy using PSS (i) before healing and (ii) after healing. Reprinted with permission from Ref. [84]. 2017, Nature Research; and (**d**) schematic diagram of ozone treatment. Reprinted with permission from Ref. [137]. 2017, Nature Research.

**Figure 16 nanomaterials-14-00410-f016:**
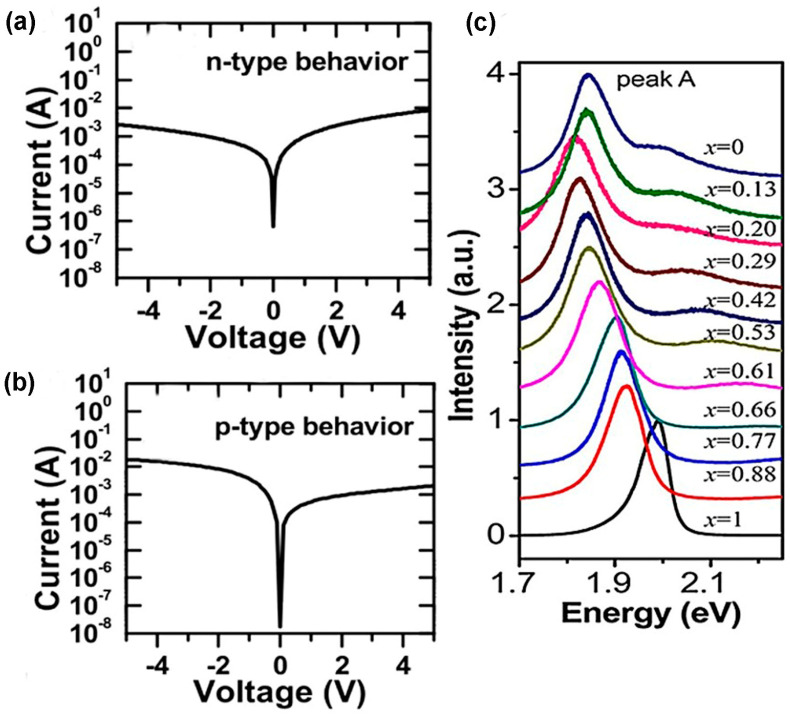
(**a**) N-type behavior of MoS_2_; (**b**) p-type behavior of MoS_2_. Reprinted with permission from Ref. [140]. 2014, American Chemical Society; and (**c**) the band gap variation in MoS_2_ according to doping. Reprinted with permission from Ref. [56]. 2013, American Chemical Society.

**Figure 17 nanomaterials-14-00410-f017:**
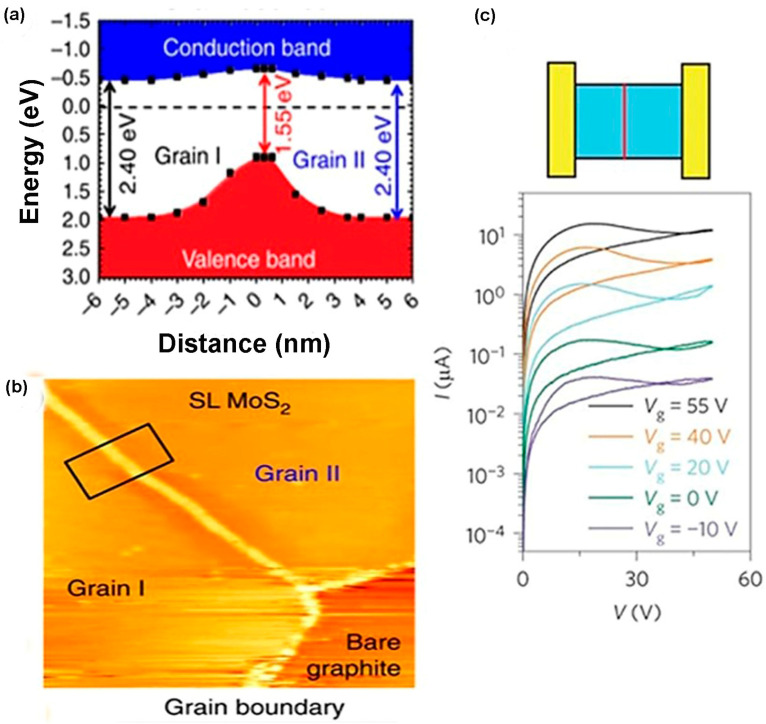
(**a**) The schematic diagram of band gap variation in adjacent grain boundaries; (**b**) the STEM images of MoS_2_ grain boundaries that show the band gap variation. Reprinted with permission from Ref. [142]. 2015, Nature Communications; and (**c**) the TMDC-based memristor and its output characteristics. Reprinted with permission from Ref. [37]. 2013, American Physical Society.

**Figure 18 nanomaterials-14-00410-f018:**
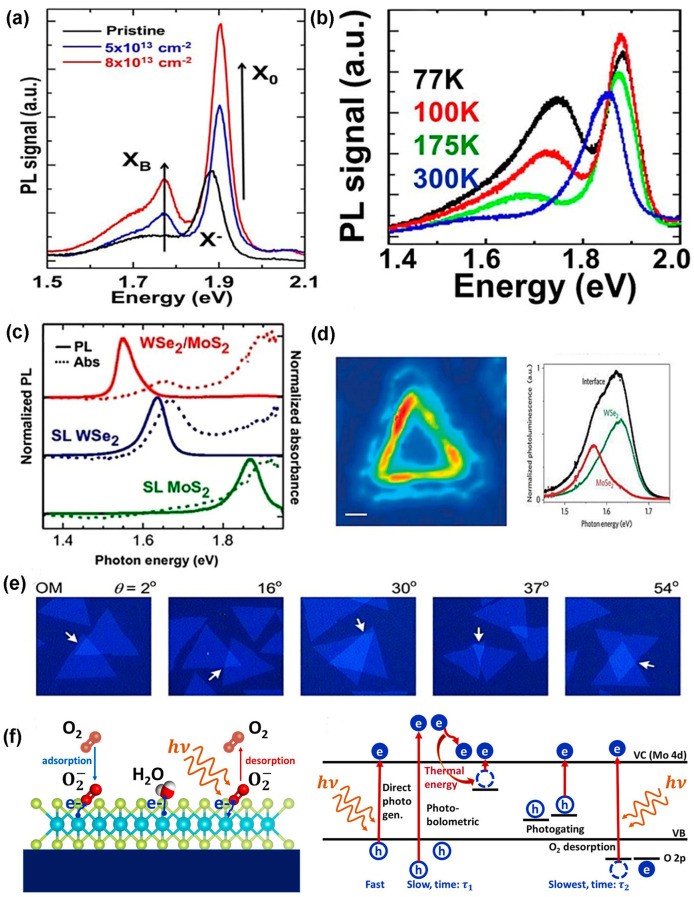
(**a**) The band gap shifting of MoS_2_ after ion irradiation; (**b**) the PL spectroscopy of MoS_2_ after annealing confirms the S defect formation. Reprinted with permission from Ref. [44]. 2013, Nature Research; (**c**) the band gap shifting due to heterostructure of TMDC materials; (**d**) the STEM and PL images confirm that high recombination occurred at the interface of two different layers; and (**e**) the variation in SHG intensity with the tilting angle of TMDC films. Reprinted with permission from Ref. [2]. 2016, IOPScience; (**f**) variation in photoresponse due to defects. Reprinted with permission from Ref. [49]. 2023, Elsevier.

**Figure 19 nanomaterials-14-00410-f019:**
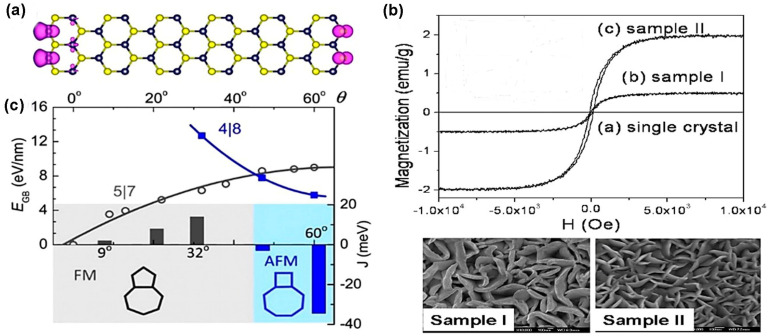
Magnetic properties of TMDC materials: (**a**) the zigzag structural model of WS_2_ edges. Reprinted with permission from Ref. [2]. 2016, IOPscience; (**b**) thickness dependent on hysteresis loop of MoS_2_. Reprinted with permission from Ref. [23]. 2007, American Chemical Society; and (**c**) variation in magnetic properties with the grain boundary. Reprinted with permission from Ref. [45]. 2013, American Chemical Society.

**Table 1 nanomaterials-14-00410-t001:** General features of various transition metal dichalcogenides.

TMDC Materials	Band Gap (Bulk) (eV)	Bandgap (Monolayer) (eV)	Polymorphs	Raman Modes for Monolayer		Dopants	
	Indirect Band Gap	Direct Band Gap		In-Plane	Out-of-Plane	N-Type	P-Type
MoS_2_	1.23 [24]	1.88 [24]	1H-MoS2 (Semiconducting), 1T-MoS2 (Metallic), 1T′-MoS2 (Semi-metallic), 2H-MoS2 (Semiconducting), 3R-MoS2 (Semiconducting) [25]	385.28 [26]	405.39 [26]	Mn, Re, Tc [1,27], F, Cl, Br, I [28] Tetracene [29]	Nb [30], Ta [27] N, P, As [28] F4TCNQ, PTCDA [29]
MoSe_2_	1.09 [31]	1.57 [31]	1H-MoSe2 (Semiconducting), 1T′-MoSe2 (Semiconducting), 1T″-MoSe2 (Semiconducting), 1T-MoSe2(Metallic) [32]	290.30 [33]	239.20 [33]	F, Cl, Br, I [34]	Nb [35], C, O, S, Te [34]
MoTe_2_	1 [36]	1.1 [36]	1T-MoTe2(Metallic), 1T′-MoTe2 (Semiconducting), 2H-MoTe2(Semiconducting) [37]	230.00 [38]	170.00 [38]	Benzyl Viologen [39]	Al_2_O_3_ [40]
WSe_2_	1.21 [31]	1.67 [31]	1T-WSe2(Metallic), 1T′-WSe2 (Semiconducting), 2H-WSe2 (Semiconducting) [41,42]	249.00 [43]	259.00 [43]	K [39]	(NH4)2S [39]
WS_2_	1.32 [31]	2.03 [31]	1T-WS2(Metallic), 1T′-WS2 (Semiconducting), 2H-WS2 (Semiconducting), 3R-WS2 (Semiconducting) [41,42,44]	350.00 [45]	420.00 [45]	Mn, Re [1] LiF, Hydrazine [39]	Nb, S [46]
WTe_2_	0.95 [47]	1.07 [48]	1T-WTe2(Metallic), 1T′-WTe2 (Semiconducting), 2H-WTe2 (Semiconducting), 3R-WS2 (Semiconducting) [41,49]	-	-	Re [50]	Ta [50]

**Table 2 nanomaterials-14-00410-t002:** Various defects and defect engineering procedures.

	Zero Dimensional Defects (0D)					One Dimensional Defects (1D)	Two Dimensional Defects (2D)
	Vacancies		Anti-Sites	Substitutions	Adatoms		
Defects	Four type of vacancies- single chalcogen vacancy, dichalcogenide vacancy, Vacancy complex of transition metal and three chalcogen atoms on one plane, Vacancy complex of transition metal and three dichalcogenide pairs [52], rotational defects [66]		Two Types—Transition metal atoms substitute chalcogen atoms, or vice versa [53]	Replacement of chalcogen or transition metal atoms by foreign atoms [53]	Four possible places of adatoms adsorption: top of the transition metal atom, top of the chalcogen atom, between or on the transition metal and chalcogen bond, void of the hexagonal plane [54]	Film surface chalcogen vacancies result in zig-zag line defects [52]. Grain boundaries form by merging lattice mismatched grains [80], with shapes influenced by factors like chalcogen or transition metal atom concentration, growth kinetics, ambient conditions, and precursor temperature [72].	Localized strain on film surface forms ripple that 2D defects impact bond length, bond angle and surface curvature [76], Van-der-Waals hetero-stacking of various TMDC flakes by deposition or transferring flakes one after another from substrate by wet-chemical process forms 2D defect due to lattice mismatch [2], Moiré patterns of TMDC flakes form 2D defects [2]
Defect Engineering Procedures	In-situ defect engineering	Controlling chalcogen vapor during TMDC growth suppress point defects by minimizing chalcogen vacancies that has high vapor pressure [53]	Reducing anti-site defects by controlling chalcogen vacancies [78]	Tunning substitutional point defects by loading appropriate initial reactants [2,81]	Controlling adatoms into growth environment of TMDC materials promote sulfurization or, selenization and increase film continuity [81]	Tunning edge formation or line defects by controlling sulfur vapor in lateral direction during growth [9], Maintaining shape and size of grain boundaries by controlling reaction chamber pressure [2]	Tunning 2D heterostructure interface by controlling growth temperature [2]
	Ex-situ defect engineering	Electron beam or focused ion beam irradiation creates vacancies [82], Ar plasma makes chalcogen vacancy [2], (3-mercaptopropyl) trimethoxy silane (MPS) [83] and poly (4-styrenesulfonate) (PSS) minimize sulfur vacancies [84], Ozon and laser rays in presence of oxygen suppresses chalcogen vacancies [83,84], helium ion creates point defects [85]		Ion beam creates substitutional defects [2,82], Oxygen plasma repairs chalcogen vacancies by introducing foreign oxygen atoms [85]	Ion beam creates adatom impurities at the vacancy sites [2,82]	Plasma irradiation creates line vacancies by migrating and agglomerating chalcogen vacancies [2]	Ripple formation and separation of TMDC layers are performed using CHF3 and SF6 plasma treatment [86]

**Table 3 nanomaterials-14-00410-t003:** Electrical, optical, and magnetic properties of TMDC materials.

Properties	MoS_2_	MoSe_2_	MoTe_2_	WS_2_	WSe_2_	ReS_2_	ReSe_2_
On/Off ratio	8 × 10^8^ [39]	1 × 10^6^ [39]	1 × 10^6^ [39]	1 × 10^7^ [39]	1 × 10^7^ [39]	1 × 10^6^ [6]	1 × 10^5^ [143]
Mobility (cm^2^V^−1^s^−1^)	60 [144]	12 [145]	0.9 [38]	486 [146]	100 [147]	23.1 [6]	5 [143]
Dielectric Constant	2.6-2.9 [148]	10.37 [149]	2.1-7 [150]	Not found	Not found	Not found	Not found
Absorption wavelength range(nm)	400-700 [151]	900 [152]	1115 [153]	420-700 [154]	Not found	Not found	1560 nm [155]
Magnetic properties	Non-magnetic [80,156]	Non-magnetic [80]	Non-magnetic [80]	Non-magnetic [80]	Non-magnetic [80]	Non-magnetic [80]	Non-magnetic [80]

## Data Availability

The data presented in this study are available on request from the corresponding author. The data are not publicly available due to academic restrictions.

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
