# Peer review of "Defects and Defect Engineering of Two-Dimensional Transition Metal Dichalcogenide (2D TMDC) Materials"

_nanomaterials, 2024, doi:10.3390/nano14050410_

Round 1

Reviewer 1 Report

Comments and Suggestions for Authors

The work provides a general review on defect and defects engineering of 2D TMDC nanomaterials. The work needs significant improvement to be considered for publication in MDPI Nanomaterials, followed by the comments.

As there are plenty of reports on the same topic in existing literature, even with specific defects, properties, and applications [https://doi.org/10.34133/2019/4641739, https://doi.org/10.1039/D2CS00931E, https://doi.org/10.1007/s12274-022-5016-9, https://doi.org/10.1016/S1872-2067(21)63945-1, https://doi.org/10.1088/1674-4926/40/7/070403, https://doi.org/10.1021/acsnano.0c09666, https://doi.org/10.1002/admi.202000494]. How does this simple general introduction write with a touch to all the general aspects without any significant addition to the knowledge of readers stands out with the previous reports.

Authors need to reduce the extensive lengths, and provide specific content related to the recent reports.

Introduction needs more information from defects and their implications. It is just like a general introduction to 2D materials. More references on 2D materials such as BP, ReS2, ReS2 etc. should be included.

Line 47, it should be black phosphorus instead of black phosphor.

MOCVD related references should be added in line 94-97. Does the reference provide results related to all the synthesis techniques mentioned. [https://doi.org/10.1557/adv.2018.237, https://doi.org/10.1016/j.carbon.2022.10.037]

Section 3 and section 4 should be replaced with each other, first classification of the defects, and then defect characterizations.

Line 226, add – and can also give rise to additional secondary phases with the increasing laser power.

Table 1 should be described in the text.

The quality of the images needs to be improved, kindly take the copy rights link to reproduce the original figures.

Abbreviations are repeated such as for CVT, PVD, PL etc. Introduce only one time.

The role of intrinsic and extrinsic defects on the slowdown of the optoelectronic properties due to charge trapping centers should be discussed. [https://doi.org/10.1016/j.mtnano.2023.100382]

Line 980, Re- is a n-type dopant… However, more recent experimental work on ReS2, ReSe2 for pressure and temperature effects electronic and optoelectronic applications should be discussed and cited.

The role of interfacial defects should be introduced and the impact on electronic properties should be discussed more in detail.

The work lacks quantitative results for electronic, optical, and magnetic properties.

Mainly the works related to MoS2 are discussed, another material such with Se as well as Te should also be considered for comparing the impact of defects on their physical, electronic, optical, and magnetic properties.

Lacking relevant reference for sentences are missing such as lines 1108-1111.

Tables should be presented with quantitative results for comparison.

References should be in proper MDPI Nanomaterials format.

Author Response

At the outset, we would like to thank you the reviewer for the thorough review of our submitted manuscript. Please find attached the revised manuscript

Thank you

Shyam Aravamudhan

Reviewer 2 Report

Comments and Suggestions for Authors

In the presented work entitled „Defects and Defects Engineering of 2D TMDC Materials” by M.F. Hossen et al., the Authors presents review of different types of defects (named as zero-dimensional (0D), one-dimensional (1D), and two-dimensional (2D)) that can be present in 2D transition metal dichalcogenides (TMDC) materials. The existing defect engineering methods that relate to both formation, and reduction of defects are discussed. Finally, an attempt is made to correlate the impact of defects and the properties of the TMDC materials. 

The presented review is generally well-written and has clear structure, so the contained information is accessible to the readers. The subject itself is timely and important, however the references are relatively dated. This is important since several previous review in this subject already exist in reputable journals and cover the same period of time (see Chem. Soc. Rev. 47 (2018) 3100-3128 or (2D Mater. 3 (2016) 022002). Hence, I cannot recommend this review for publication in its present form. The Authors must include much more recent references (only several references from 2020 and above are currently present) and extend their review by new results in comparison to the existing review studies. It should be also clearly said what is new about the present review that cannot be found in the previous ones, and why publishing it right now is important. The review articles are specific form of paper and should be very strict in this regard. There are only few minor aspects that should be corrected:

1.     The Authors mention in multiple places that defects can be used to engineer interfaces of TMDS with other materials. However, I am missing one crucial aspect related to the potential barrier engineering at the TMDS-metal interface. Can such barrier be modulated (lowered) by proper defect engineering? For more details on potential barriers at the TMDS-metal interface please refer to (ACS Appl. Mater. Interfaces 7 (2015) 25709−25715 and Physical Review B 97 (2018) 195315).

2.     The text in all tables (1-3) is very small and hard to read. All the tables are also included as a relatively low-quality pictures instead of a text. Please provide the table in a text format with font size corresponding to the main text.

3.     The Authors should provide the full name of transition metal dichalcogenides before introducing their abbreviations. The abbreviations should not be present in the title.

After addressing all above corrections, I am willing to reconsider this paper for publication.

Author Response

(The authors gave the same response as above.)

Round 2

Reviewer 1 Report

Comments and Suggestions for Authors

Authors have addressed all the comments and significantly revised the manuscript. Reviewer recommend the manuscript for publication in the present form.

Reviewer 2 Report

Comments and Suggestions for Authors

The Authors addressed most of my critique. However, this review is still presenting limited novelty when comparing to the existing studies. Hence, the present paper can be published in the Nanomaterials in its present form but it will not provide significant contribution to the field in my opinion.